# Quantifying the Accuracy–Interpretability Trade-Off in Concept-Based Sidechannel Models

## Abstract

Concept Bottleneck Models (CBNMs) are deep learning models that provide interpretability by enforcing a bottleneck layer where predictions are based exclusively on human-understandable concepts. However, this constraint also restricts information flow and often results in reduced predictive accuracy. Concept Sidechannel Models (CSMs) address this limitation by introducing a sidechannel that bypasses the bottleneck and carry additional task-relevant information. While this improves accuracy, it simultaneously compromises interpretability, as predictions may rely on uninterpretable representations transmitted through sidechannels. Currently, there exists no principled technique to control this fundamental trade-off. In this paper, we close this gap. First, we present a unified probabilistic concept sidechannel meta-model that subsumes existing CSMs as special cases. Building on this framework, we introduce the Sidechannel Independence Score (SIS), a metric that quantifies a CSM's reliance on its sidechannel by contrasting predictions made with and without sidechannel information. We propose SIS regularization, which explicitly penalizes sidechannel reliance to improve interpretability. Finally, we analyze how the expressivity of the predictor and the reliance of sidechannel jointly shape interpretability, revealing inherent trade-offs across different CSM architectures. Empirical results show that state-of-the-art CSMs, when trained solely for accuracy, exhibit low representation interpretability, and that SIS regularization substantially improves their interpretability, intervenability, and the quality of learned interpretable task predictors. Our work provides both theoretical and practical tools for developing CSMs that balance accuracy and interpretability in a principled manner.

## 1 Introduction

Concept-based models (CBMs) have emerged as a promising direction for interpretable deep learning (Poeta et al., 2023; Koh et al., 2020; Espinosa Zarlenga et al., 2022; Mahinpei et al., 2021). CBMs achieve interpretability by incorporating high-level, human-understandable *concepts* explicitly within their architecture. A well-known example CBM is the Concept Bottleneck Model (Koh et al., 2020) (CBNM), which first predicts concepts (e.g. *whiskers*, *tail*) using a neural network, and then maps these predicted concepts to the target task using a linear layer (e.g. *cat*). This architecture makes predictions inherently explainable: one can trace decisions back to predicted concepts (e.g. 'the model predicted *cat* because it sees *whiskers* and *a tail*'), and the linear layer provides additional transparency by revealing the influence of each concept on the final prediction.

Despite their appeal, early CBMs such as CBNMs suffered from a significant drop in task accuracy compared to black-box models. This is because concepts in CBNMs form an *information bottleneck*: the model must rely solely on concepts to perform the task. In practice, it is often infeasible to design a concept set that fully captures all the information needed for accurate predictions, which makes a performance gap inevitable.

To address this limitation, recent works have augmented CBMs with an additional *sidechannel* that transmits extra information beyond concepts (Mahinpei et al., 2021; Espinosa Zarlenga et al., 2022;

Barbiero et al., 2023; Sawada & Nakamura, 2022).[1] This sidechannel can take different forms, such as embeddings (Yuksekgonul et al., 2022) or one-hot masks (Debot et al., 2024), depending on the CBM in question. By leveraging this sidechannel, CBMs can close the accuracy gap with black-box models, often achieving similar performance irrespective of the used concept set.

However, this gain in accuracy comes at the cost of some interpretability. Unlike in CBNMs, predictions in sidechannel CBMs are no longer solely determined by concepts, but also by uninterpretable units of informations (e.g. an embedding). The more a model relies on information captured by its sidechannel, the more accurate but less interpretable it becomes. In the extreme, the task predictor could rely completely on the sidechannel, bypassing the concepts entirely.

While some works have acknowledged this tension in a limited way by designing techniques that try to maximize concept usage (Kalampalikis et al., 2025; Havasi et al., 2022; Shang et al., 2024), we show that this is a weaker criterion than minimizing sidechannel reliance. The interpretability cost is never quantified or explicitly optimized, and most evaluations and training objectives still focus on accuracy, leaving two key gaps in the literature: the lack of a *metric* to measure interpretability cost in sidechannel CBMs, and the lack of a principled way to *optimize* for this cost during training.

In this work, we aim to close these gaps. Our contributions are the following:

1. **Unified view:** We show that existing sidechannel CBMs can all be interpreted as different parametrizations of a single *meta-model*, which we formalize as a high-level probabilistic graphical model (PGM).

2. **Interpretability metric:** Using this PGM, we provide a natural evaluation method for sidechannel CBMs by equipping them with a *bottleneck mode*, where predictions only depend on the concepts. Based on bottleneck mode, we subsequently introduce the *Sidechannel Independence Score* (SIS), which quantifies this interpretability cost.

3. **Training objective:** We propose *SIS regularization*, a method that explicitly optimizes sidechannel CBMs for interpretability by penalizing reliance on the sidechannel.

4. **Interpretability discussion:** We discuss how different notions of interpretability in CSMs arise when considering both the expressivity of the predictor and its reliance on the sidechannel.

Through contributions (1), (2) and (4), we provide a clearer understanding of the structure and trade-offs of sidechannel CBMs. Through (3), we introduce practical ways for developing models that are not only accurate but also interpretable.

## 2 BACKGROUND

**Concept Bottleneck Models.** Concept Bottleneck Models (CBNMs) (Koh et al., 2020) are CBMs that consist of two functions: a *concept predictor* ($X \rightarrow C$) that maps some low-level input features $X$ (e.g. an image) to high-level, human-understandable concepts $C$ (e.g. *whiskers*, *tail*), and a *task predictor* ($C \rightarrow Y$) that maps the concepts to some target task $Y$ (e.g. *cat*). CBNMs are trained by directly supervising both concepts and task with the goal of aligning each concept to a human interpretation and obtaining high task performance. This supervision comes either from concept and task labels in the dataset or from vision-language models, the latter removing the need for expensive human annotations (Oikarinen et al., 2023). Typically, CBNMs use a neural network as concept predictor and a linear layer or neural network as task predictor. A key downside of CBNMs is that their accuracy for predicting the task $Y$ is limited by the employed concepts $C$, as they form an information bottleneck.

**Concept Sidechannel Models.** Concept Sidechannel Models (CSMs) are CBMs that address the information bottleneck issue by predicting $Y$ not only using $C$ but also using some additional information (Espinosa Zarlenga et al., 2022; Sawada & Nakamura, 2022; Yuksekgonul et al., 2022; Barbiero et al., 2023). This additional information comes in different forms for different CSMs. Some examples of this are an embedding predicted from $X$ (Mahinpei et al., 2021) and a one-hot mask predicted from $X$ (Debot et al., 2024). The central idea behind CSMs is primarily to achieve (near) black-box accuracy, and secondarily to be as interpretable as possible.

---

[1]In the literature, this sidechannel is also often referred to as a "residual" or "sidepath".

**Notation.** We write random variables in upper case (e.g. $p(X)$) and their assignments in lower case (e.g. $p(X = x)$). When it is clear from the context, we will abbreviate assignments (e.g. $p(x)$ means $p(X = x)$). Conditional distributions are written using a vertical bar (e.g. $p(Y|X)$ or $p(Y = y|X = x)$).

# 3 METHOD

In this section, we present our proposed method. We begin by clarifying the distinction between representation interpretability, which refers to the interpretability of the model's intermediate representations, and functional interpretability, which refers to the interpretability of the prediction process (Section 3.1). Subsequently, we introduce a probabilistic *CSM meta-model*, which provides a unified framework for representing all CSMs (Section 3.2). We then demonstrate how this meta-model enables two modes of inference: the *default* mode and the *bottleneck* mode, in which the sidechannel is deactivated (Section 3.3). The delineation of these two modes naturally motivates the introduction of a novel metric, the Sidechannel Independence Score (SIS), which quantifies the distance between the modes and, consequently, the dependence of the original CSM on its sidechannel (Section 3.4). We then show how this distance can be employed as a regularization criterion for CSMs, providing explicit control over the accuracy–interpretability trade-off in CSMs. Finally, in Section 3.5, we discuss how the sidechannel not only supplements task-relevant information missing in the concepts, but also enhances concept-to-task expressivity for some CSMs, highlighting a trade-off between representation interpretability and functional interpretability.

## 3.1 REPRESENTATION INTERPRETABILITY VS FUNCTIONAL INTERPRETABILITY

Based on existing informal notions within the CBM community (Barbiero et al., 2025), we make two distinctions in interpretability for CBMs: *representation interpretability* and *functional interpretability*. We first consider the classical form of interpretability used in machine learning.

**Definition 3.1** (Functional Interpretability of CBMs)**.** *A CBM is* functionally interpretable *if and only if the mapping from the CBM's concepts (and sidechannel) to the task is an interpretable function.*

What one considers an interpretable function is subjective to the human user. According to Rudin et al. (2022), standard cases of interpretable functions include linear layers, logic rules and small decision trees. For CSMs, only considering this form of interpretability is insufficient as CSMs also make their task prediction using uninterpretable representations (i.e. the sidechannel). Therefore, we must also consider a form of interpretability that depends on the use of such uninterpretable representations: *representation interpretability*.

**Definition 3.2** (Representation Interpretability of CBMs)**.** *A prediction is said to be* representation interpretable *if and only if it is derived exclusively from units of information that are themselves interpretable. A concept-based model is* more *representation interpretable* if a larger fraction of its task-level predictions are representation-interpretable.

We make the following assumption regarding interpretable units:

**Assumption 3.1** (Interpretable Units of Information in CBMs)**.** *In a CBM, the concepts $C$ constitute the only interpretable units, since they are explicitly supervised to align with some human understanding.[2] All other internal variables are uninterpretable, irrespective of their form, as they lack explicit human alignment (e.g. sidechannels).*

**Proposition 3.1** (Representation Interpretability of CBNMs and CSMs)**.** *Under Assumption 3.1:*

1. *A CBNM is fully representation-interpretable irrespective of the used task predictor (e.g. linear layer or neural network): every prediction $\hat{y}$ depends only on the concepts.*

2. *A CSM is partially representation-interpretable: a prediction $\hat{y}$ is representation-interpretable if and only if $\hat{y}$ does not depend on the sidechannel.*

---

[2]This assumes appropriate measures have been taken to prevent concept leakage (Marconato et al., 2022), which may otherwise misalign the concepts.

**Example 3.1.** *Consider a CSM with a single sidechannel neuron and a linear layer mapping the concepts and neuron to the task. If the class label $y$ is directly encoded in the sidechannel neuron, the model can predict $y$ by relying solely on this neuron. This model is not representation interpretable (since predictions only use the sidechannel) but functionally interpretable (since the linear layer clearly shows how $y$ is obtained). Conversely, if the sidechannel neuron carries no information (a dummy value), but the model uses a neural network on the concepts and neuron, then predictions are representation interpretable (since they depend only on concepts) but not functionally interpretable (due to the black-box predictor).*

Current CSMs focus on achieving functional interpretability while maintaining high accuracy, and do not consider representation interpretability, despite its importance as illustrated above. **Therefore, our focus lies on evaluating and improving representation interpretability in CSMs.**

### 3.2 THE META-MODEL OF CSMS

A wide range of CSMs have been developed in recent years. While they differ in many ways, their high-level structure is very similar. These models can be understood as different instantiations of a common underlying high-level model, which we call the *CSM meta-model*. This CSM meta-model consists of three functions: a concept predictor $\phi_c : X \to C$, a sidechannel predictor $\phi_z : X \to Z$, and a task predictor $\phi_y : C, Z \to Y$. Both the functions $(\phi_c, \phi_z, \phi_y)$ and the variables $(C, Z, Y)$ can differ across CSMs. For example, concepts may be continuous ($C = \mathbb{R}^{n_C}$) or discrete ($C \in \{0, 1\}^{n_C}$). Sidechannels also vary in form, with examples including embeddings ($Z \in \mathbb{R}^{|Z|}$) (Mahinpei et al., 2021) and a one-hot mask ($Z \in \{0, 1\}^{|Z|}$) (Debot et al., 2024).

Without loss of generality, the meta-model can be expressed as a probabilistic graphical model (PGM) (Figure 1). The PGM factorizes the joint distribution as:

$$p(y, c, z, x) = p(x) \cdot p(c|x) \cdot p(z|x) \cdot p(y|c, z) \qquad (1)$$

with general task inference defined as

$$p(y|x) = \sum_{c,z} p(c|x) \cdot p(z|x) \cdot p(y|c, z) \qquad (2)$$

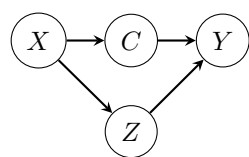

Figure 1: CSM Meta-model

Here, the summation is taken over all possible assignments of the concepts $c$ and the sidechannel $z$. For continuous variables, the summations are replaced with integrals.

Specific CSMs then correspond to particular parametrizations of this PGM. Even models without explicit probabilistic semantics can be cast into this framework (such as Concept Embedding Models (Espinosa Zarlenga et al., 2022)).[3] Two examples that we will work out in detail are CRM and CMR.

**Example 3.2** (Concept Residual Model (CRM) (Mahinpei et al., 2021))**.** *In CRM, concepts are represented as delta distributions: $p(c|x) = \delta(\hat{c} - c)$ where $\hat{c} = g(x)$ with $g$ a neural network. The sidechannel $z$ is an embedding also represented by a delta distribution, i.e. $p(z|x) = \delta(\hat{z} - z)$ where $\hat{z} = f(x)$ with $f$ a neural network. The task predictor $p(y|c, z)$ is a neural network. Due to the sifting property of the Dirac delta, the inference expression simplifies to a single evaluation of the task predictor: $p(y|x) = p(y|\hat{c}, \hat{z})$. CRM is not functionally interpretable due to the neural task predictor.*

**Example 3.3** (Concept Memory Reasoner (CMR) (Debot et al., 2024))**.** *In CMR, concepts are modelled as Bernoulli random variables: $p(c|x) = g(x)$ with $g$ a neural network with sigmoid activation. The sidechannel $z$ is a categorical distribution representing a one-hot mask: $p(z|x) = f(x)$ with $f$ a neural network with softmax activation. The task predictor $p(y|c, z)$ possesses a set of learned logic rules. At inference time, it evaluates only the rule indicated by the sidechannel $z$ on the concepts. The resulting inference expression is: $p(y|x) = \sum_{c \in \{0,1\}^{n_C}} \sum_{z=1}^{n_R} p(c|x) \cdot p(z|x) \cdot p(y|c, z)$, where $n_R$ is a hyperparameter signalling CMR's number of learned rules. CMR is functionally interpretable due to its rule-based task predictor.*

In Appendix A, we explain additionally for the following CSMs how they parametrize our meta-model: Concept Embedding Models (Espinosa Zarlenga et al., 2022), Concept Bottleneck Models

---

[3]Concepts and sidechannels can be represented as delta distributions, so that the summation in Equation 2 reduces to a single evaluation due to the Dirac delta's sifting property.

with Additional Unsupervised Concepts (Sawada & Nakamura, 2022), Hybrid Post-Hoc Concept Bottleneck Models (Yuksekgonul et al., 2022), and Deep Concept Reasoner (Barbiero et al., 2023).

### 3.3 DEFAULT AND BOTTLENECK MODES FOR CSMs

Consider a CSM where the sidechannel $z$ is replaced with a value independent of the input $x$. In this case, the task prediction depends only on $x$ through the concepts, i.e. $p(y|c, x) = p(y|c)$. This effectively disables the sidechannel, ensuring that predictions are fully representation interpretable. This can be achieved by replacing the distribution $p(z|x)$ with a distribution $p(z)$.

We can therefore define two operation modes of a CSM:

$$\textbf{Default Mode:} \qquad p_{\theta,\phi,\psi}(y|x) = \sum_{c,z} p_\theta(c|x) \cdot \boxed{p_\phi(z|x)} \cdot p_\psi(y|c,z) \qquad (3)$$

$$\textbf{Bottleneck Mode:} \qquad p_{\theta,\gamma,\psi}(\bar{y}|x) = \sum_{c,z} p_\theta(c|x) \cdot \boxed{p_\gamma(z)} \cdot p_\psi(y|c,z) \qquad (4)$$

where the subscripts $\{\theta, \phi, \psi, \gamma\}$ indicate learnable parameters. Notice that the two modes differ *exclusively* on whether their predictions use $p_\phi(z|x)$ or $p_\gamma(z)$, We can always make predictions using either mode, with bottleneck mode guaranteeing full representation interpretability. Note that default mode corresponds to the standard inference of CSMs (Equation 2).

A natural question is how to obtain $p(z)$. A simple approach is to marginalize over the input distribution and approximate this using the training dataset $\mathcal{D}$, since the true data distribution $p(X)$ is typically unknown: $p_\gamma(z) = \sum_x p(x) \cdot p_\phi(z|x) \approx \frac{1}{|\mathcal{D}|} \sum_{x \in \mathcal{D}} p_\phi(z|x)$. Here, bottleneck mode has the same parameters as default mode (so $\gamma \equiv \phi$). We will discuss an alternative approach in Appendix B where instead a prior $p(z)$ is explicitly learned.

### 3.4 MEASURING AND OPTIMIZING FOR REPRESENTATION INTERPRETABILITY

The definition of the two modes defined in the previous section allows use to devise a metric to measure how often a CSM relies on its sidechannel by checking the agreement between predictions from default and bottleneck mode.

**Definition 3.3** (Sidechannel Independence Score - SIS). *Let $y_x$ and $\bar{y}_x$ denote predictions from default and bottleneck mode, respectively, obtained by thresholding the corresponding distributions $p(y|x)$ and $p(\bar{y}|x)$ at a given threshold probability. We define the Sidechannel Independence Score (SIS) as:*

$$SIS = \mathbb{E}_{x \sim p(X)}[\mathbb{1}[y_x = \bar{y}_x]]$$

The SIS measures the frequency with which the model's prediction changes when the sidechannel is removed, i.e. how dependent it is on the sidechannel rather than concepts. In practice, the SIS cannot be computed exactly since the true data distribution $p(X)$ is unknown. Instead, we approximate it empirically using a dataset $\mathcal{D}$: $\widehat{SIS} = \frac{1}{|\mathcal{D}|} \sum_{x \in \mathcal{D}} \mathbb{1}[y_x = \bar{y}_x]$. Assuming the dataset is i.i.d., we can use Hoeffding's inequality to provide formal guarantees about the model's SIS (and thus representation interpretability) on unseen data. In particular, $p(|\widehat{SIS} - SIS| \geq \epsilon) \leq 2e^{-2|\mathcal{D}|\epsilon^2}$. For instance, if we find an empirical $\widehat{SIS} = 60\%$ on a test set with size $|\mathcal{D}| = 1000$, then the $95\%$ confidence interval of $SIS$ is $[58\%, 62\%]$. This metric is pragmatic in the sense that a human can easily interpret it and decide on whether they consider the model representation-interpretable enough to trust it.

Most CSMs are currently trained to maximize accuracy (see Section 4). Our meta-model enables explicit optimization for representation interpretability by introducing a loss term that penalizes discrepancies between predictions from default mode ($p(y|x)$) and bottleneck mode ($p(\bar{y}|x)$). Any suitable divergence, such as total variation distance or symmetric Kullback-Leibler divergence, can be used. This approach integrates seamlessly into any CSM. For example, when maximizing the likelihood of training data, the objective becomes:

$$\arg\max_{\phi,\psi,\gamma,\theta} \left[ \sum_{(x,c,y)\in\mathcal{D}} (\log p_{\phi,\psi}(y|c,x) + \alpha \cdot \log p_\theta(c|x) - \beta \cdot \text{DIV}(p_{\phi,\psi}(y|c,x)||p_{\gamma,\psi}(\bar{y}|c,x))) \right]$$

where $\alpha$ and $\beta$ are hyperparameters, and $\text{DIV}(\cdot||\cdot)$ denotes a chosen divergence measure. We refer to this additional term as **SIS regularization**. It can also be incorporated into alternative training schemes, such as sequential or joint CBM training (Koh et al., 2020).

Furthermore, instead of marginalizing over the input $x$ to obtain a prior $p(z)$, one can introduce a learnable prior. This reduces computational cost, simplifies optimization, and in some cases increases the expressivity of the model in bottleneck mode. For more details, we refer to Appendix B. Importantly, this method is applicable to any CSM, regardless of its parametrization within our meta-model PGM (e.g. choice of sidechannel $z$ or form of $p(y|c, z)$).

### 3.5 EXPRESSIVITY OF CSMs IN BOTTLENECK MODE

In default mode, CSMs are typically universal classifiers: they are as expressive as neural networks for classification. This universality comes from the sidechannel, which allows them to learn any mapping from input to task ($X \rightarrow Y$), irrespective of the employed concept set.

When in bottleneck mode, however, their ability to learn input-to-task mappings is limited by the quality of the concept set. Because the concept set is dependent on the dataset, a general comparison between CSMs cannot be made. Instead, we can analyze their expressivity in learning mappings from concepts to task ($C \rightarrow Y$). Some CSMs are more expressive than others in this regard, for instance:

- CRM (Mahinpei et al., 2021) applies a neural network to the concepts. This is fully expressive but not functionally interpretable.

- CBM-AUC (Sawada & Nakamura, 2022) applies a linear layer to the concepts, which is functionally interpretable but not expressive.

- CMR (Debot et al., 2024) falls in between, as it applies a set of learned logic rules to the concepts. Its expressivity depends on the number of learned rules, and increases with this capacity.

For more examples, we refer to Appendix A.

This difference in $C \rightarrow Y$ expressivity between default and bottleneck mode for some CSMs implies that they do not only use the sidepath to use information related to $Y$ that cannot be found in $C$ but also to improve their $C \rightarrow Y$ expressivity.

This yields two important insights. First, for achieving the same accuracy on expressive tasks $Y$, a functionally interpretable but inexpressive CSM (e.g. CBM-AUC) must rely more heavily on its sidechannel than a non-functionally interpretable but expressive CSM (e.g. CRM). Thus, the former may achieve higher functional interpretability but lower representation interpretability than the latter. Second, even if the concepts are sufficient for the task (i.e. a perfect predictor exists given a sufficiently expressive model), a functionally interpretable but inexpressive CSM will still need to use its sidechannel to achieve high accuracy. We show this empirically in Section 5, and illustrate it with the following simple example.

**Example 3.4.** *Consider two binary concepts $c_1$ and $c_2$ and a task defined as their logical XOR ($y := c_1 \oplus c_2$). Suppose a CSM uses a linear layer to map the concepts $c$ and a sidechannel $z$ (some neurons) to the task $y$. In bottleneck mode, the CSM cannot predict $y$ accurately, since it is not linearly separable. In default mode, the model can encode features such as $c_1 \wedge \neg c_2$ and $\neg c_1 \wedge c_2$ into the sidechannel. The linear layer can then compute the logical OR of these neurons, thereby solving the XOR task. In this way, the sidechannel effectively extends the concept bottleneck with combinations of concepts to increase the model's expressivity.*

## 4 RELATED WORK

While explicitly measuring and optimizing representation interpretability has not been studied directly, several related research directions pursue overlapping but distinct goals. Most of these works focus on encouraging models to use concepts extensively, whereas our focus lies in minimizing reliance on the sidechannel. Note that the latter entails the former, but goes even further.

A major line of research in CBMs emphasizes *intervenability* (Espinosa Zarlenga et al., 2022; 2023; Havasi et al., 2022): when concept predictions are replaced with their ground truth, downstream task accuracy should improve as much as possible. Such interventions are designed to simulate human expert interaction at decision time. However, intervenability is influenced by many factors, for instance: (1) the extent to which concepts are used, (2) whether interventions remain in-distribution, and (3) architectural design choices. Of these, only (1) partially relates to representation interpretability. Importantly, a model may achieve high intervenability by heavily relying on concepts while still encoding substantial task-relevant information in the sidechannel. In this case, intervenability remains high even though representation interpretability may be low.

Some works attempt to *disentangle* the sidechannel from the concepts (Zabounidis et al., 2023): make it complementary to the concepts, not re-encode the same information. This approach improves representation interpretability to some degree by preventing concept-related information from being in the sidechannel, but does not reduce sidechannel usage beyond this. Moreover, enforcing disentanglement substantially harms accuracy in CSMs that rely on sidechannels for expressivity (Barbiero et al., 2023; Debot et al., 2024; Sawada & Nakamura, 2022), even though disentanglement is not strictly necessary for achieving representation interpretability (see Appendix C).

Other works introduce methods to maximize *concept utilization* during prediction (Kalampalikis et al., 2025; Shang et al., 2024). These approaches encourage reliance on concepts but do not directly address sidechannel reduction (see Section 5). Furthermore, they are tailored to CSMs with embedding-based sidechannels, leaving open how they might generalize to alternative architectures. Finally, they may require structural modifications to the model (e.g. a factorizable task predictor (Shang et al., 2024)), whereas our approach does not require this. Similarly, Zhang et al. (2024) introduce an embedding-based CSM with a factorized task predictor and a regularization approach, which *does* encourage sidechannel reduction, but is specific to their task predictor.

Finally, Havasi et al. (2022) propose a metric known as the *completeness score*, which estimates how fully concepts capture predictive information for the task. This is computed by learning a distribution $q(z|c)$ after training and relies on mutual information. While informative, it suffers from two limitations. First, mutual-information–based quantities are less intuitive for humans than accuracy-based metrics like SIS. Second, it is not a measure for representation interpretability, as it may be high even though representation interpretability is poor (see Appendix C for an in-depth discussion and a comparison between their $q(z|c)$ and our $p(z)$). Zhang et al. (2024) also introduce a metric for interpretability, but this can be only be computed for factorizable task predictors and similarly does not measure representation interpretability.

## 5 EXPERIMENTS

### 5.1 SETUP

This section provides essential information about the experiments (see Appendix D for details). We report results averaged over three seeds, and we give standard-deviations as shaded areas. For pareto curves, we only give the results for the first seed (see Appendix D for the remaining ones).

**Datasets.** We use three standard concept-based datasets: CelebA (Liu et al., 2015), with 200k celebrity faces annotated with facial attribute concepts (e.g. *blond hair*, *beard*); CUB (Welinder et al., 2010), where the task is to classify birds; and MNIST-Addition (Manhaeve et al., 2018), where the task is to predict the sum of two digit images. CelebA's concepts are insufficient for the task; MNIST-Addition's are sufficient but has an expressive task. CUB's concepts are sufficient, has an inexpressive task, but has difficult concept prediction. For the results on CUB, we refer to Appendix D.2.

**Models.** We use the following CSMs in our experiments: Concept Residual Models (CRM) (Mahinpei et al., 2021), Concept Embedding Models (CEM) (Espinosa Zarlenga et al., 2022), Deep Concept Reasoner (DCR) (Barbiero et al., 2023), and Concept Memory Reasoner (CMR) (Debot et al., 2024). We also define Linear Residual Model (LRM) as CRM but with a linear layer as task predictor. DCR, CMR, and LRM are *functionally interpretable* but *inexpressive* in bottleneck mode (see Section 3.5); CRM and CEM are *not functionally interpretable* but *expressive* in bottleneck mode. As **baselines**, we consider approaches that maximize concept usage (Shang et al., 2024; Kalampalikis et al., 2025).

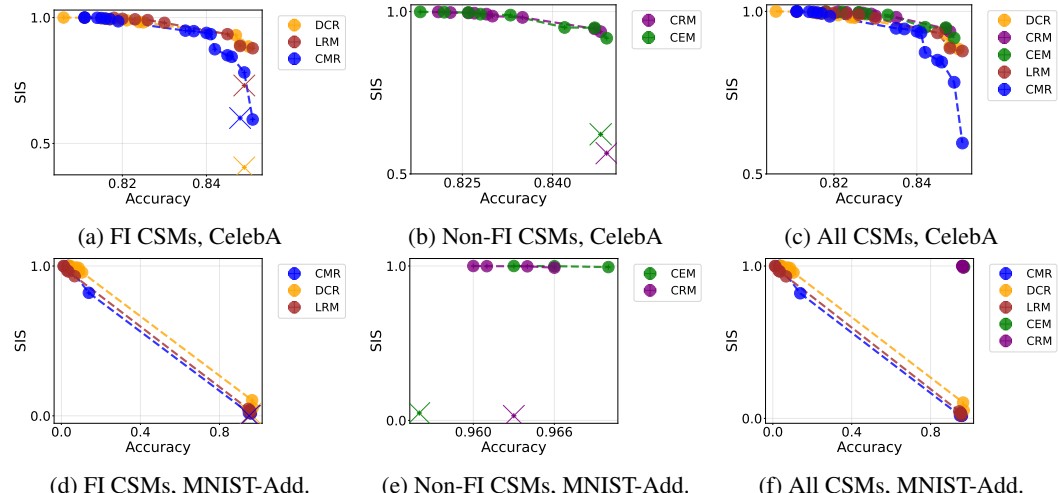

(a) FI CSMs, CelebA  (b) Non-FI CSMs, CelebA  (c) All CSMs, CelebA

(d) FI CSMs, MNIST-Add.  (e) Non-FI CSMs, MNIST-Add.  (f) All CSMs, MNIST-Add.

Figure 2: Accuracy vs representation interpretability trade-off in CSMs. Each point is a hyperparameter configuration, keeping only pareto-efficient points. Crosses are the most accurate configuration trained without SIS regularization (not in (c) and (f)). FI means "functionally interpretable".

## 5.2 RESULTS AND DISCUSSION

We consider the following research questions: **(Interpretability)** How representation interpretable are state-of-the-art CSMs? How much does SIS regularization improve their representation interpretability? What is the trade-off w.r.t. accuracy? How do current methods encouraging concept usage for CRM-like models (Section 4) fare regarding representation interpretability? **(Side-effects)** Does SIS regularization improve intervenability? Does SIS regularization improve the quality of learned interpretable task predictors?

**Current optimization of CSMs results in uninterpretable models (Figure 2, crosses), contrary to when using SIS regularization (Figure 2, circles).** Models that are optimized only for accuracy have a low SIS score, meaning they significantly use the sidechannel. Notably, this is even the case when the sidechannel is completely unnecessary, e.g. when training expressive CSMs on datasets where the concepts are sufficient for the task (Figure 2e). Conversely, when using our SIS regularization, CSMs become significantly more representation interpretable, and the human can choose how much accuracy to trade for how much interpretability.

**When using the sidechannel is unnecessary, SIS regularization ensures CSMs avoid it (Figure 2e).** With sufficient concept sets, expressive CSMs can reach high accuracy without relying on the sidechannel, effectively functioning as concept bottleneck models with the same interpretability.

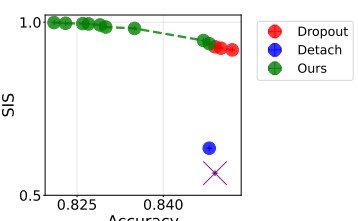

Figure 3: Accuracy vs SIS for CRM on CelebA, comparing with concept usage approaches. The cross is the most accurate CRM trained without any approach.

**Inexpressive CSMs require the sidechannel for expressive tasks (Figure 2d).** Despite MNIST-Addition having sufficient concepts, the linear expressivity of these CSMs' bottleneck mode makes them either very accurate but completely not representation interpretable, or completely representation interpretable but very inaccurate. Interestingly, our analysis shows that CMR seems unable to exploit its non-linear expressivity. In Appendix D, we address this by replacing its rule learner with an existing alternative, enabling near-perfect accuracy and interpretability on this task.

**Existing concept usage approaches yield smaller gains in representation interpretability (Figure 3).** While approaches that encourage higher concept usage for CRM-like CSMs also increase representation interpretability, their improvements are smaller than those achieved with SIS regular-

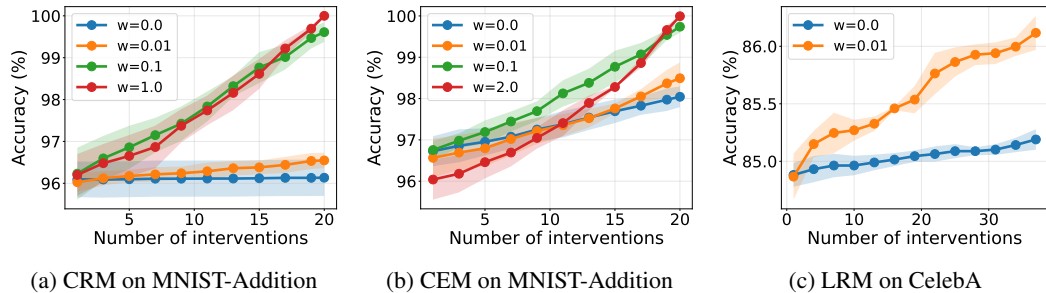

(a) CRM on MNIST-Addition      (b) CEM on MNIST-Addition      (c) LRM on CelebA

Figure 4: Intervenability in CSMs with ($w > 0$) and without ($w = 0$) SIS regularization for different regularization weights $w$. The y-axis denotes accuracy after intervening on a number of concepts denoted by the x-axis.

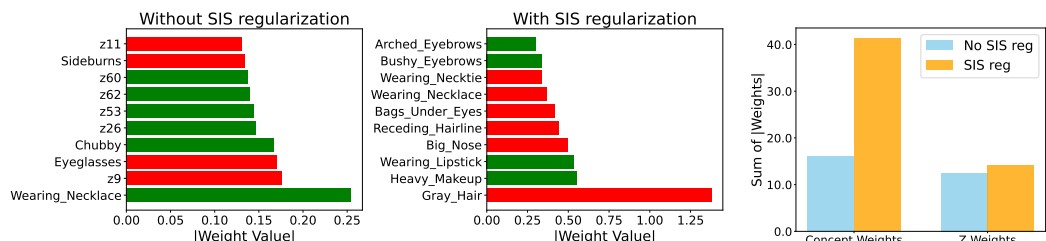

(a) Top 10 largest weights. Bars represent absolute weight magnitudes, with green and red indicating positive and negative weights, respectively. Weights corresponding to the sidechannel's neurons are denoted with a 'z'.

(b) Total absolute weight magnitudes assigned to concepts and sidechannel neurons.

Figure 5: Inspection of LRM's linear layer for predicting the task 'Young' for CelebA, comparing LRMs trained with and without SIS regularization.

ization. We observe that *dropout* (Kalampalikis et al., 2025) enhances SIS only insofar as accuracy is not compromised, and that *detach* (Shang et al., 2024) only slightly improves SIS.

**SIS regularization increases intervenability (Figure 4).** As SIS regularization reduces sidechannel reliance in the CSM, an automatic side-effect and advantage is that the CSM relies more on the concepts to predict the task, making it more responsive to concept interventions. For more curves, see Appendix D.

**SIS regularization improves the task predictor quality of functionally interpretable CSMs (Figure 5).** We compare the most accurate LRM configurations trained with and without regularization on CelebA (84.75% vs 84.94% accuracy). Without SIS regularization, many of the most contributing weights of LRM's linear task predictor are uninterpretable latent factors from the sidechannel (e.g. *z11*), harming interpretability. In contrast, with SIS regularization, large weights correspond to semantically meaningful concepts (e.g. *Gray Hair*), while reliance on the sidechannel is suppressed (Figure 5a), and the total weight magnitude shifts significantly toward concepts rather than the sidechannel (Figure 5b).

## 6 CONCLUSION

We addressed a fundamental gap in concept-based models: the lack of principled methods to measure and control the trade-off between accuracy and interpretability in concept sidechannel models (CSMs). We proposed a unified probabilistic meta-model that places existing CSMs within a single framework, enabling us to disconnect representation interpretability from functional interpretability. Building on this, we introduced the Sidechannel Independence Score (SIS) as a metric that quantifies a model's representation interpretability. We demonstrated how SIS can serve as a regularization objective, allowing human users to explicitly control the extent to which models rely on uninterpretable sidechannels. Our experiments reveal that state-of-the-art CSMs, when trained using

typical objectives, are not genuinely interpretable, and that SIS regularization produces models that are more representation interpretable, more responsive to interventions and more transparent in their task predictors. Our analysis also allowed us to find a weakness in a state-of-the-art CSM (CMR), being unable to exploit its theoretical expressivity, which we addressed. Our contributions provide both theoretical foundations and practical tools for developing interpretable CSMs.

**Limitations and future work** Future work could investigate the accuracy-interpretability trade-off among datasets beyond vision (e.g. language) and more CSMs, and investigate the effect of SIS regularization on the quality of learned rules for rule-based CSMs.

**Reproducibility statement.** All our experiments are seeded, and we will make the code publicly available upon publication of the paper. Moreover, in Appendix D, we describe in detail the setup of each experiment, the implementation of each model, and the training setup.

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

# Supplementary Material

TABLE OF CONTENTS

# A  OVERVIEW OF STATE-OF-THE-ART CSMs

*In this section, we give more examples of how state-of-the-art CSMs are parametrizations of our meta-model.*

We do not give extensive details about the models, only a high-level overview. We also take the examples CRM and CMR again as they appear in the main text.

**Concept Residual Model (CRM) (Mahinpei et al., 2021).** In CRM, concepts are represented as delta distributions: $p(c|x) = \delta(\hat{c} - c)$ where $\hat{c} = g(x)$ with $g$ a neural network. The sidechannel $z$ is an embedding also represented by a delta distribution, i.e. $p(z|x) = \delta(\hat{z} - z)$ where $\hat{z} = f(x)$ with $f$ a neural network. The task predictor $p(y|c, z)$ is a neural network. Due to the sifting property of the Dirac delta, the inference expression simplifies to a single evaluation of the task predictor: $p(y|x) = p(y|\hat{c}, \hat{z})$. CRM is not functionally interpretable due to the neural task predictor.

**Concept Embedding Models (CEM) (Espinosa Zarlenga et al., 2022).** In CEM, concepts are represented as delta distributions: $p(c|x) = \delta(\hat{c} - c)$ where $\hat{c} = g(x)$ with $g$ a neural network. The sidechannel $z$ consists of 2 embeddings per concept, each represented by a delta distribution, i.e. for $i \in \{1..n_C\}$ and $j \in \{1, 2\}$, $p(z_{i,j}|x) = \delta(\hat{z}_{i,j} - z_{i,j})$ where $\hat{z}_{i,j} = f_{ij}(x)$ with $f_{ij}$ a neural network. The task predictor $p(y|c, z)$ first mixes the two embeddings of each concept using the concept scores ($z_i = c_i \cdot z_{i,1} + (1 - c_i) \cdot z_{i,2}$), and the resulting embeddings are concatenated and fed through a neural network. Due to the sifting property of the Dirac delta, the inference expression simplifies to a single evaluation of the task predictor: $p(y|x) = p(y|\hat{c}, \hat{z})$. CEM is not functionally interpretable due to the neural task predictor.

**Concept Memory Reasoner (CMR) (Debot et al., 2024).** In CMR, concepts are modelled as Bernoulli random variables: $p(c|x) = g(x)$ with $g$ a neural network with sigmoid activation. The sidechannel $z$ is a categorical distribution representing a one-hot mask: $p(z|x) = f(x)$ with $f$ a neural network with softmax activation. The task predictor $p(y|c, z)$ encodes a set of learned logic rules. At inference time, it evaluates only the rule indicated by the sidechannel $z$ on the concepts. The resulting inference expression is: $p(y|x) = \sum_{c \in \{0,1\}^{n_C}} \sum_{z=1}^{n_R} p(c|x) \cdot p(z|x) \cdot p(y|c, z)$, where $n_R$ is a hyperparameter signalling CMR's number of learned rules. CMR is functionally interpretable due to the rule-based task predictor.

**Deep Concept Reasoner (DCR) (Barbiero et al., 2023).** In DCR, concepts are modelled as delta distributions: $p(c|x) = \delta(\hat{c} - c)$ where $\hat{c} = g(x)$ with $g$ a neural network. The sidechannel $z$ represents a fuzzy logic rule, given by 2 fuzzy values per concept, signalling the relevance and polarity of each concept in the rule: for $i \in \{1..n_C\}$ and $j \in \{1, 2\}$, $p(z_{i,j}|x) = \delta(\hat{z}_{i,j} - z_{i,j})$ where $\hat{z}_{i,j} = f_{ij}(x)$ with $f_{ij}$ a neural network with sigmoid activation. The task predictor $p(y|c, z)$ uses fuzzy logical inference to deduce the task from the fuzzy logic rule and the predicted concepts. DCR is functionally interpretable due to the rule-based task predictor.

**Concept Bottleneck Model with Additional Unsupervised Concepts (CBM-AUC) (Sawada & Nakamura, 2022).** In CBM-AUC, concepts are also delta distributions. The sidechannel consists of two components: some unsupervised concepts and a weight for each concept and unsupervised concept. Both are represented by delta distributions. The task predictor computes the dot product of all concepts and weights. CBM-AUC is functionally interpretable due to the linear task predictor.

**Hybrid Post-hoc Concept Bottleneck Models (PCBM-h) (Yuksekgonul et al., 2022).** In PCBM-h, concepts are delta distributions, and the sidechannel is a single value denoted by a delta distribution. The task predictor applies some interpretable predictor (e.g. linear layer) to the concepts, providing a score $y_1$, which is summed with the value predicted by the sidechannel. PCBM-h is functionally interpretable due to the interpretable predictor followed by a simple sum.

**Expressivity.** Of these models, CRM and CEM are universal classifiers ($C \rightarrow Y$) in bottleneck mode, but not functionally interpretable. DCR and CBM-AUC are functionally interpretable, but have a linear expressivity ($C \rightarrow Y$) in bottleneck mode. PCBM-h is functionally interpretable but its expressivity depends on the used predictor. CMR is functionally interpretable and its expressivity in bottleneck mode is limited only by the number of rules it uses.

## B  LEARNABLE PRIOR $p(z)$

When using SIS regularization, we propose to use a learnable prior for $p(z)$ as opposed to a marginalization approach. For some models, avoiding the marginalization can improve expressivity. For instance, for CMR, using the marginalization approach means that the final task prediction is still using a single logic rule, with uncertainty modelled over which rule to use. As a consequence, CMR's expressivity ($C \rightarrow Y$) would still be linear. By instead allowing CMR to use the entire set of learned rules (which would be impossible by the marginalization approach, as this value for $z$ is out-of-distribution), CMR's expressivity becomes bounded only by the number of rules it has learned.

This ensures the SIS regularization is computationally quite cheap, this ensures that the SIS regularization requires only a single forward pass of the model and computing a KL divergence between the original forward pass and this additional one.

## C  LIMITS OF DISENTANGLEMENT AND CONCEPT COMPLETENESS SCORE

### C.1  DISENTANGLING FOR REPRESENTATION INTERPRETABILITY

*In this section, we argue that disentangling the sidechannel from the concepts is neither a sufficient, nor necessary condition for achieving a representation-interpretable CSM. Moreover, it may even needlessly harm the CSM's accuracy. This highlights that representation interpretability (and SIS) capture(s) a distinct notion of interpretability not already addressed by disentanglement.*

Zabounidis et al. (2023) propose disentangling the sidechannel from the concepts. This improves representation interpretability by preventing concept-related information from leaking into the sidechannel. However, as noted in Section 4, this is not sufficient: task-relevant information that is unrelated to the concepts is still free to appear in the sidechannel.

Importantly, disentanglement is not strictly necessary for a CSM to be representation interpretable. The simplest example is when the task predictor relies only on the concepts for each prediction. In this case, the prediction is representation interpretable regardless of what information the sidechannel encodes.

A more interesting example is the following, where disentanglement is clearly unnecessary and even hurts the model's accuracy. Suppose the dataset is such that the concepts are sufficient for solving the task. That is, they capture all task-relevant information. Consider CMR, a CSM where the sidechannel is a one-hot mask. The task predictor contains a set of logic rules. In default mode, the predictor uses the sidechannel to select a single rule to apply to the concepts; in bottleneck mode, it applies all rules simultaneously. For some inputs, the prediction made in default mode (via rule selection) may coincide with the prediction in bottleneck mode (via all rules), meaning the prediction is representation interpretable. However, if the sidechannel were disentangled from the concepts, rule selection could not exploit any task-relevant information (since all of it resides in the concepts). In that case, accuracy would collapse, as the rule selection would effectively become random.

### C.2  CONCEPT COMPLETENESS SCORE VS SIDECHANNEL INDEPENDENCE SCORE

*In this section, we argue that the concept completeness score (Havasi et al., 2022) does not capture representation interpretability, comparing it to our SIS metric.*

The *concept completeness score* (CCS) $\tau$ is defined by Havasi et al. (2022) as

$$\tau = \frac{I(y; c)}{I(y; c, x)} \approx \frac{H(y) + \mathbb{E}_{(x,c)\sim\mathcal{D}}[\log \mathbb{E}_{z\sim q(z|c)}[p(y|c, z)]]}{H(y) + \mathbb{E}_{(x,c)\sim\mathcal{D}}[\log \mathbb{E}_{z\sim p(z|x)}[p(y|c, z)]]} \tag{5}$$

where $H(\cdot)$ denotes entropy. This score is used to estimate how much task-relevant information is present in the concepts. Note that the numerator uses $q(z|c)$: CCS will be high whenever predictions based on $z$ derived from $c$ resemble those based on $z$ derived from $x$. Crucially, this can happen even if the task predictor heavily relies on the sidechannel, so long as all the information in the sidechannel is also present in the concepts. Intuitively: *the concept completeness score measures*

*how much information more than concepts is encoded in the sidechannel; SIS measures how much the sidechannel is used.* CCS can be very high for a completely uninterpretable prediction, which is not the case for SIS. Concept-related information encoded in an uninterpretable form (e.g. embeddings) is not interpretable, since humans cannot identify what it represents, despite CCS being high. Furthermore, concepts are explicitly supervised to align with human-understandable interpretations, whereas sidechannels are not. Thus, even if a sidechannel has a simple form (e.g. a logic rule, as in DCR (Barbiero et al., 2023)) and is predicted from the same information as the concepts, we argue it remains uninterpretable. Consider the following two examples.

Consider an example with DCR. Suppose the concepts capture all task-relevant information, and the model uses five concepts $c_1, c_2, \ldots, c_5$. For an input $x$ with ground-truth label $y = $ True, assume all concepts are predicted as *True*, and the sidechannel outputs the rule $y \leftarrow c_1 \wedge c_2 \wedge \cdots \wedge c_5$. DCR's task predictor applies this rule to the predicted concepts, correctly predicting $y = $ True. For this individual prediction, $\tau$ would be high, since the rule is derived from concept-related information. However, it is opaque to the human why the model used this particular rule. Why not $y \leftarrow c_1 \wedge c_2$, which would yield the same correct label, or even $y \leftarrow c_1$, or $y \leftarrow $ True? In the last case, the concepts are bypassed entirely, and the prediction is made solely by the sidechannel: a neural network, hence uninterpretable. The core issue is that the sidechannel is not explicitly *aligned* with human-understandable semantics, unlike the concepts. If it were, then it would be a concept.

Consider an example with LRM. Suppose the concepts capture all task-relevant information, and the model uses some concepts and a sidechannel $z$ that is a single neuron. These are passed to a linear layer for the task prediction. If the linear layer has learned every weight to be zero except the weight on the sidechannel, then the task prediction probability is determined entirely by the sidechannel, i.e. by the underlying neural network that predicts $z$. In this case, $\tau$ will be 1, since $z$ is derived from concept-related information. However, the interpretability the linear layer provides is completely lost. A human examining the weights will only see that the task decision (completely) depends on $z$, but has no way to understand how the individual concepts contribute. While the linear layer provides functional interpretability, it is useless because the model is completely not representation interpretable: the linear layer purely uses an uninterpretable representation.

Similarly, the example in Section 3.5 would also have a high completeness score.

**Using $p(z)$ versus $q(z \mid c)$.** One could consider defining bottleneck mode using $q(z \mid c)$ instead of $p(z)$, where the former is parameterized by a neural network. This substitution would increase expressivity: CSMs that currently act as linear classifiers ($C \rightarrow Y$) in bottleneck mode under $p(z)$ would become universal classifiers ($C \rightarrow Y$) under $q(z \mid c)$, since $q$ is parameterized by a neural network. However, as we explained above, this added flexibility comes at the cost of interpretability. With $p(z)$, the mapping from $C \rightarrow Y$ in bottleneck mode is interpretable for functionally interpretable CSMs, whereas with $q(z \mid c)$ a neural network is put between concepts and task predictions, reducing interpretability.

# D  EXPERIMENTS

## D.1  EXPERIMENTAL DETAILS

**Datasets.** In MNIST-Addition (Manhaeve et al., 2018), each input consists of two MNIST images each representing some digit. The task is to predict the sum of the 2 digits. The concepts denote which digit is in each image. In CelebA (Liu et al., 2015), the concepts are face attributes such as *Blond Hair* and *Wears Make-up*. As tasks, we take the concepts *Male*, *Young* and *Attractive*, dropping them from the concept set.

**Metrics.** For CelebA, we use regular accuracy and SIS as defined in the main text. For MNIST-Addition, we use *subset* versions of these metrics due to the large imbalance and mutually exclusive nature of both concepts and tasks.

**Seeds.** We use seeds 1, 2 and 3.

**Hard concepts.** To avoid the problem of concept leakage which harms interpretability (Marconato et al., 2022), we employ hard concepts (as opposed to soft concepts) by thresholding the concept predictions at 50% before passing them to the task predictor, which is common for CBMs.

**Training.** We use a training objective similar to sequential training. We maximize the likelihood of our data (see Section 3.4), with as difference that we use the predicted concepts $\hat{c}$ instead of the ground truth labels for $c$ in the task predictor $p(y|c, z)$. This makes the model more robust and aware of mistakes in the concept prediction. Importantly, we do not allow the gradient from $y$ to pass through $c$ (so we avoid the joint training objective some CBMs employ), as this is known to cause task leakage, harming interpretability (Mahinpei et al., 2021). For CEM, we also use its *randint* regularization. We always use the AdamW optimized with learning rate 0.001 and train for 80 epochs, restoring the weights that resulted in the lowest validation loss. Our training/validation split is 9/1.

**Backbones.** For each model, we use the same backbone (which differs between datasets). For CelebA, we train on pretrained ResNet18 embeddings (He et al., 2016), similar to Debot et al. (2024). Images are first resized to (224, 224) using bi-linear interpolation. They are then normalized per channel with means (0.485, 0.456, 0.406) and standard deviations (0.229, 0.224, 0.225). By removing the last layer of the pretrained ResNet18, using the resulting model on each image, and flattening the output, we obtain an embedding. For MNIST-Addition, we train directly on the images. The backbone is a CNN (learned jointly with the rest of the CSM) that consists of the following layers: a convolution layer with 6 output channels and kernel size 5, a max-pool layer with kernel size and stride 2, a ReLu activation, a convolution layer with 16 output channels and kernel size 5, a max-pool layer with kernel size and stride 2, a ReLu activation, a flattening layer, a linear layer with $emb\_size//2$ output features, and 3 linear layers each with $emb\_size//2$ output features and ReLu activation (except the last one). The backbone is applied to each MNIST image, and the two resulting embeddings are concatenated. $emb\_size$ is a hyperparameter. If any hyperparameters are unmentioned, we use the default values.

**Concrete CSMs.** We will describe our implementation of the different CSMs. For each CSM, the input first passed through the backbone before being passed to the layers we will describe in what follows. Unless explicitly mentioned otherwise, each linear layer has $emb\_size$ output features.

For CRM (Mahinpei et al., 2021), the concept predictor is a neural network consisting of 2 linear layers with ReLU activation, and a linear layer with $n_C$ output features with sigmoid activation. The task predictor is a neural network with 3 linear layers and a linear layer with $n_y$ output features and sigmoid activation. The sidechannel is a neural network with 2 linear layers with ReLU activation, and a linear layer, outputting an embedding. The prior $p(z)$ is denoted by a single learnable Torch embedding object with $emb\_size$ weights.

For CEM (Espinosa Zarlenga et al., 2022), the sidechannel is a neural network with 3 linear layers with $2 \cdot c\_emb\_size \cdot n_c$ output features and ReLU activation, and a linear layer with $2 \cdot c\_emb\_size \cdot n_c$ output features (with $c\_emb\_size$ a hyperparameter). This is reshaped to $n_c, 2, c\_emb\_size$. The concept predictor concatenates for each concept its 2 embeddings, and applies a (different) linear layer to the resulting embedding with 1 output feature and sigmoid activation. This is the concept prediction. The task predictor uses the concept predictions to mix for each concept its 2 embeddings, and concatenates the resulting embeddings. The result is passed through 3 linear layers with ReLU activation, and a linear layer with $n_y$ output features and sigmoid activation. The prior $p(z)$ is denoted by a single learnable Torch embedding object with $2 \cdot c\_emb\_size \cdot n_c$ weights.

For DCR (Barbiero et al., 2023), the sidechannel is a neural network with 3 linear layers with $2 \cdot c\_emb\_size \cdot n_c$ output features and ReLU activation, and a linear layer with $2 \cdot c\_emb\_size \cdot n_c$ output features (with $c\_emb\_size$ a hyperparameter). This is reshaped to $n_c, 2, c\_emb\_size$. The concept predictor concatenates for each concept its 2 embeddings, and applies a (different) linear layer to the resulting embedding with 1 output feature and sigmoid activation. This is the concept prediction. The task predictor uses the concept predictions to mix for each concept its 2 embeddings. Each resulting *concept embedding* is passed through a (different) neural network consisting of 3 linear layers with $c\_emb\_size$ output features and ReLU activation, and a linear layer with $2 \cdot n_y$ output features with sigmoid activation. These should be interpreted as, for each concept, its polarity and relevance for each task, as defined by DCR. These are used together with the concept predictions to infer the task prediction. We employ DCR's logic formula (see Barbiero et al. (2023)) using the product t-norm. The prior $p(z)$ is denoted by a single learnable Torch embedding object with $2 \cdot c\_emb\_size \cdot n_c$ weights. To avoid needing to finetune DCR's *temperature* hyperparameter, which a human user would do to find a preferred rule parsimony, we modelled DCR's role and relevance with a sigmoid activation instead of their rescaled softmax activation.

For CMR (Debot et al., 2024), the concept predictor is a neural network consisting of 2 linear layers with ReLU activation, and a linear layer with $n_C$ output features with sigmoid activation. The sidechannel is a neural network with 3 linear layers with ReLU activation, and a linear layer with $n_r \cdot n_y$ output features, where $n_r$ is CMR's allowed number of rules to learn (hyperparameter). This is reshaped to $(n_y, n_r)$ with a softmax on the last dimension (as this is a categorical random variable). The task predictor consists of a Torch embedding object with shape $n_y, n_r, rule\_emb\_size$ weights, with $rule\_emb\_size$ a hyperparameter. This is CMR's rulebook in its latent representation, which is decoded into explicit logic rules by using a neural network on each rule embedding with 3 linear layers with ReLU activation and a linear layer with $3 \cdot n_c$ output features with softmax activation. The task predictor then uses CMR's inference formula (see (Debot et al., 2024)) to derive the task prediction from the categorical distribution over rules (given by the sidechannel) and the learned set of rules. In bottleneck mode, CMR applies the entire set of rules to the concepts, each making a $y$ prediction. The final prediction is the logical OR of the individual predictions. CMR's hyperparameter for its prototype regularization is set to 0 for the same reason as why we changed DCR's role and relevance activation, as this hyperparameter would be used by the human to find a preferred rule parsimony.

For LRM, we use the same setup as for CRM, with as difference that the task predictor is a linear layer with $n_y$ output features and sigmoid activation.

**Training with and without learnable prior.** Whenever we use SIS regularization, we train with a learnable prior. Otherwise, we compute $p(z)$ by marginalizing out $x$, as proposed in the main text. For deterministic CSMs that use an embedding as sidechannel (e.g. CRM), this typically means $p(z)$ becomes a mixture of delta distributions, with one delta per training instance (as embeddings rarely perfectly coincide). As this does not scale, we instead relax this by instead considering a single $z$, namely the average of all such embeddings.

**CMR\*.** We define CMR\* as an adaptation of CMR by equipping it with rules learned by a decision tree. Specifically, we extract the rules decision trees learned on ground truth $(c, y)$ pairs predicting positive classes. We start with decision trees with a maximum depth of 1, increasing it until the decision tree has more than 17 such rules, or depth exceeds 40. Then, we take the rules from the tree with the highest validation accuracy. We inject these rules into CMR, and allow it to learn 3 additional rules on its own. This is done through rule interventions, which CMR supports (see Debot et al. (2024)).

**For obtaining Figure 2**, we perform a **grid search** where $emb\_size$ is taken from $\{64, 128, 256\}$ and the weight of the SIS regularization is taken from $\{0, 0.0001, 0.001, 0.01, 0.05, 0.1, 0.5, 1.0, 1.5, 2.0, 3.0, 4.0, 5.0\}$. For CEM and DCR, we use $p_{randint} = 0.05$. **For obtaining Figure 4**, we have 2 phases. In the first phase, we perform the same grid search except that we do not employ SIS regularization. For each CSM, we take the most accurate configuration on the validation set. Then, in the second phase, for each CSM, we train this configuration with different values of the SIS regularization weight, taken from $\{0, 0.0001, 0.001, 0.01, 0.05, 0.1, 0.5, 1.0, 1.5, 2.0, 3.0, 4.0, 5.0\}$. For each such trained model, we compute its intervenability curve by (1) generating a random concept intervention order, (2) intervening on increasingly more concepts following this order, (3) each time computing the accuracy on the tasks after the intervention. **For obtaining Figure 5**, we take the most accurate LRM on the validation set (taken from the first mentioned grid search) with and without SIS regularization and inspect the learned weights. **For obtaining Figure 3**, we re-use the results of CRM from the first mentioned grid search, and additionally train a *dropout* version inspired by Kalampalikis et al. (2025) and a *detach* version inspired by Shang et al. (2024). For the dropout version, we extend the grid by considering values for $p_{dropout}$ from $\{0.0, 0.2, 0.4, 0.8, 0.8, 1.0\}$. For more details of these two versions, see below. **For obtaining Figure 10**, we first finetune a ResNet18 on CUB, where we add a linear layer with a softmax for predicting the classes and a linear layer with a sigmoid for predicting the concepts. We use a learning rate of 0.01 (SGD optimizer, momentum of 0.9), batch size 128 and train for 200 epochs. This is similar to the configuration many other CBM works use (Espinosa Zarlenga et al., 2022). We then drop the classification layers and run the CUB images through the ResNet18 to obtain image embeddings, which we use to train the CSMs on (instead of on the images). We use the following hyperparameter grid: SIS weight within $\{0, 0.0001, 0.001, 0.01, 0.05, 0.1, 0.5, 1.0, 1.5, 2.0, 3.0, 4.0, 5.0, 7.0, 10.0, 12.0, 20.0, 50.0\}$, $emb\_size$ 512, sidechannel number of hidden layers between 0 and 2, concept predictor number of hidden layers either 0 or 1, task predictor number of hidden layers between 0 and 2.

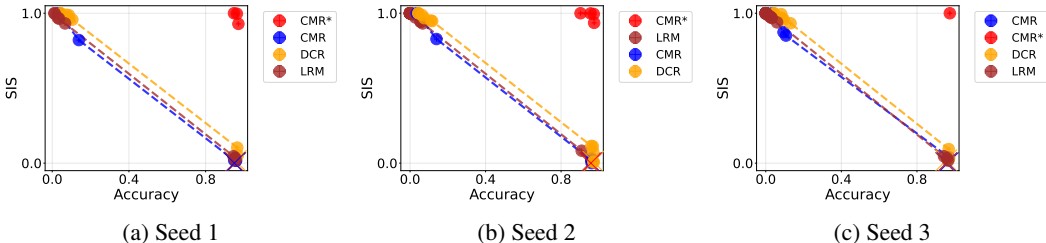

Figure 6: Accuracy vs representation interpretability trade-off in functionally interpretable CSMs, including CMR*. Different points are different hyperparameter configurations, keeping only pareto-efficient points. Crosses denote the most accurate configuration trained without SIS regularization.

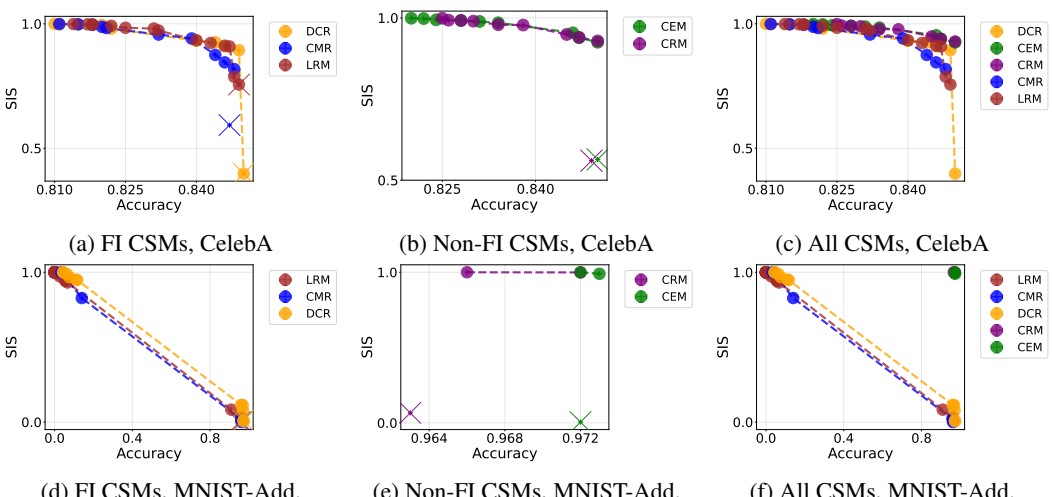

Figure 7: Accuracy vs representation interpretability trade-off in CSMs (seed 2). Different points are different hyperparameter configurations, keeping only pareto-efficient points. Crosses denote the most accurate configuration trained without SIS regularization (excluded from (c) and (f)). "(Non)-FI" denotes the figure only contains (non)-functionally interpretable CSMs.

**Detach version (CRM).** We adapt CRM's architecture to resemble Shang et al. (2024). Concretely, we redefine the task predictor as $y = f(c) + g(z)$. For $g$, we use a linear layer with $n_y$ output features and sigmoid activation. For $f$, we use a neural network consisting of 3 linear layers with ReLU activation, and a linear layer with $n_y$ output features and sigmoid activation. The model is trained by optimizing (1) a cross entropy between $f(c)$ and the label, and (2) a cross entropy between $f(c).detach() + g(x)$ and the label. The first objective tries to maximize concept usage, while the second one trains the sidechannel.

**Dropout version (CRM).** During training, for each batch, we set the entire sidechannel to 0 with probability $p_{dropout}$, which is a hyperparameter. This encourages the task predictor to rely on the concepts when the sidechannel is dropped out.

## D.2 ADDITIONAL RESULTS

With a different rule learner, CMR can achieve high accuracy when allowed to learn enough rules (Figure 6). CMR* achieves near-perfect accuracy and representation interpretability on MNIST-Addition.

Figures 7 and 8 give additional accuracy-interpretability trade-off results for seed 2 and 3. Figure 9 shows some additional intervenability curves, which also show that SIS regularization improves intervenability (similar to Figure 4 in the main text).

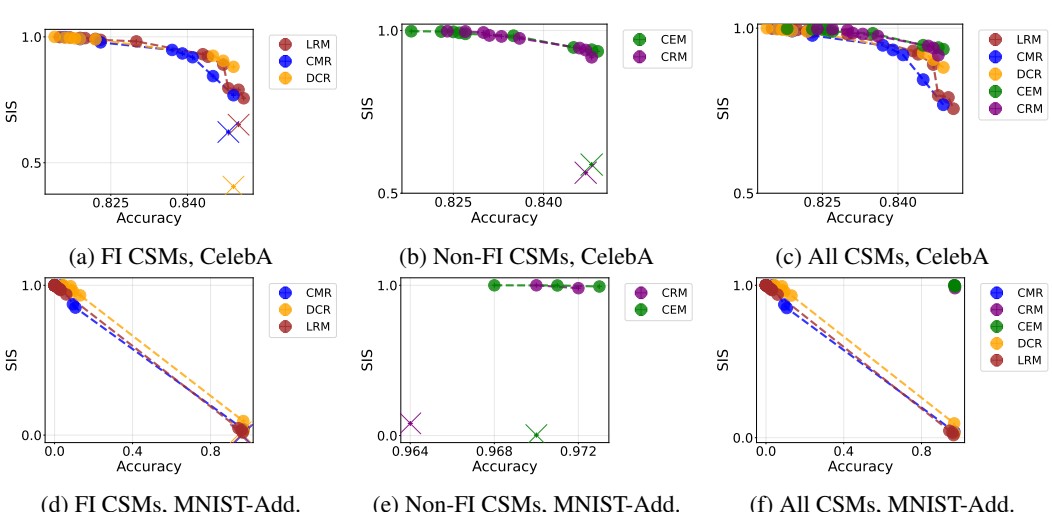

Figure 8: Accuracy vs representation interpretability trade-off in CSMs (seed 3). Different points are different hyperparameter configurations, keeping only pareto-efficient points. Crosses denote the most accurate configuration trained without SIS regularization (excluded from (c) and (f)). "(Non)-FI" denotes the figure only contains (non)-functionally interpretable CSMs.

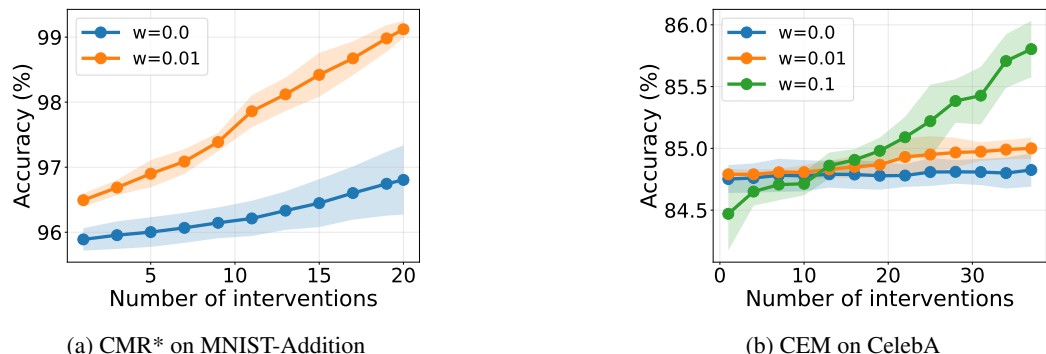

Figure 9: Intervenability in CSMs with ($w > 0$) and without ($w = 0$) SIS regularization for different regularization weights $w$. The y-axis denotes accuracy after intervening on a number of concepts denoted by the x-axis.

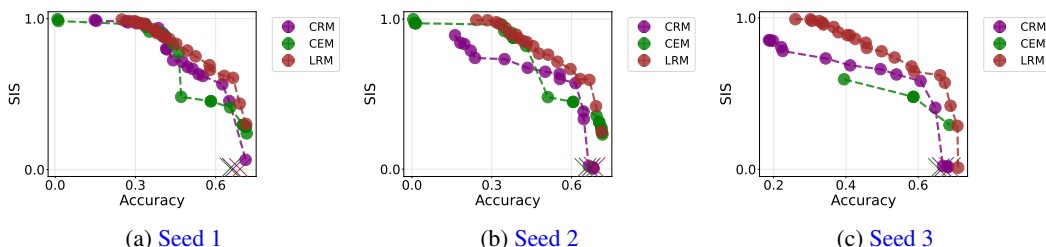

(a) Seed 1        (b) Seed 2        (c) Seed 3

Figure 10: Accuracy vs representation interpretability trade-off on CUB. Different points are different hyperparameter configurations, keeping only pareto-efficient points. Crosses denote the most accurate configuration trained without SIS regularization.

On CUB, we only used CRM, LRM and CEM because CMR and DCR do not naturally support multiclass classification. Figure 10 shows their accuracy-interpretability trade-off, which is similar to the ones in the main text.

## E   LLM USAGE DECLARATION

During writing, Large Language Models (LLMs) were used only to polish and improve the clarity of the text.

## F   CODE AND LICENSES

Our code will be made publicly available upon acceptance under the Apache license, Version 2.0. We used Python 3.10.12 and the following libraries: PyTorch v2.5.1 (BSD license) (Paszke et al., 2019), PyTorch-Lightning v2.5.0 (Apache license 2.0), scikit-learn v1.5.2 (BSD license) (Pedregosa et al., 2011), PyC v0.0.11 (Apache license 2.0). We used CUDA v12.7 and plots were made using Matplotlib (BSD license). The CelebA dataset is available for non-commercial research purposes only[4] and MNIST is available on the web with the CC BY-SA 3.0 DEED license.

---

[4]https://mmlab.ie.cuhk.edu.hk/projects/CelebA.html

