# OpenReview forum: "Quantifying the Accuracy-Interpretability Trade-Off in Concept-Based Sidechannel Models"
_ICLR.cc/2026/Conference — Submitted to ICLR 2026_

### Official Review · Reviewer_XPHp · 2025-10-22

**Soundness:** 3
**Presentation:** 4
**Contribution:** 2
**Rating:** 4
**Confidence:** 3

**Summary:**

This paper proposes a framework for quantifying and controlling the interpretability–accuracy trade-off in Concept Sidechannel Models (CSMs). The authors introduce the Sidechannel Independence Score (SIS), a metric designed to evaluate how strongly a CSM relies on its sidechannel, and propose SIS regularization, a training scheme that explicitly penalizes such reliance to enhance interpretability. Building on a unified probabilistic meta-model that encompasses existing CSM variants, the work provides a deeper understanding of how sidechannel dependence and model expressivity jointly influence interpretability.

**Strengths:**

- The paper is well-structured and presents a comprehensive probabilistic framework that unifies various CBM variants in a clear and accessible manner.
- Its motivation is both timely and significant, tackling the crucial issue of uncertainty quantification in deep models, which remains central to the pursuit of reliable and trustworthy AI.

**Weaknesses:**

- While the definitions effectively cover multiple cases, the proposed metric lacks sufficient theoretical guarantees and appears somewhat limited in novelty.
- The empirical evaluation is rather limited, as the experiments are conducted on a narrow set of datasets.

**Questions:**

1. In Figure 4(b), it is unclear why applying stronger regularization does not lead to better performance when the number of interventions is below 10. This trend is also evident when comparing regularization weights of 0.01 and 0.0, where performance actually degrades. It would be helpful to clarify whether this effect is consistent across different CBM variants. Similarly, in Figure 9(b) of the appendix, the case with $w = 0.1$ shows worse results under a low number of interventions, raising questions about the robustness of the proposed approach.

2. In relation to this, the evaluation appears to be based on a rather small set of datasets. Extending the experiments to well-established CBM benchmarks such as CUB or AWA2 could help verify the robustness of the method.

3. If I understand correctly, the CSM model introduces a side channel to enhance task accuracy through reasoning steps that are not easily captured by concepts alone, thereby sacrificing some interpretability. In other words, it seems to trade interpretability for higher accuracy. Given this, I am not sure why one would intentionally reduce accuracy to regain interpretability in such a model — wouldn’t it be more straightforward to simply use a standard CBM instead? This seems especially relevant since there appears to be an inherent accuracy–interpretability trade-off, even when using SIS regularization.

4. In connection with the previous point, could you clarify what benefits are gained from applying SIS regularization?

5. In line 240, “use” should be corrected to “us”.

---

> ### Author Response · Authors · 2025-11-21
> **Rebuttal**
>
> We thank the reviewer for taking the time to read our paper, and are glad the reviewer considers the paper well-structured, clear and accessible, and its motivation both timely and significant, tackling crucial issues in deep models.
>
> &nbsp;
>
> > The experiments are conducted on a rather small set of datasets. Extending the experiments to well-established CBM benchmarks such as CUB or AWA2 could help verify the robustness of the method.
>
> **We evaluate a considerable number of CSMs ([1-5]) on two standard concept-based datasets that together are representative for most concept-based datasets:** CelebA is a setting where concepts are insufficient for perfect prediction, but has an inexpressive task, while MNIST-Addition has sufficient concepts, but an expressive task. These datasets allow us to show many phenomena regarding the use of CSMs in these two prominent settings (e.g. the use of the sidechannel to increase expressivity).
>
> **Following your suggestion, we've now added a third dataset (CUB) which confirms our results.** As CUB's task is multiclass classification, we cannot use CMR and DCR; only CRM, LRM and CEM. We attached the results in the appendix.
>
> &nbsp;
>
> > While the definitions effectively cover multiple cases, the proposed metric lacks sufficient theoretical guarantees and appears somewhat limited in novelty.
>
> **The proposed SIS metric _does_ provide theoretical guarantees on the CSM's representation interpretability** when combined with e.g. Hoeffding's inequality (line 253) in the form of confidence intervals. These guarantees denote how interpretable the CSM is on unseen data from the same distribution.
>
> **We kindly ask the reviewer to explain why they believe the metric to be limited in novelty.** We are not aware of other works that reduce its novelty. SIS is the only existing metric that measures the cost in representation interpretability due to sidechannel usage in any CSM, irrespective of the sidechannel's and task predictor's parametrization. Currently, humans that design new CSMs or use existing ones are effectively blind to this interpretability cost.
>
> &nbsp;
>
> > In Figure 4(b), it is unclear why applying stronger regularization does not lead to better performance when the number of interventions is below 10. It would be helpful to clarify whether this effect is consistent across different CBM variants. Similarly, in Figure 9(b), the case with w=0.1 shows worse results [than w=0] under a low number of interventions.
>
> **As we explained in lines 402-407 and Figure 2, adding SIS regularization may harm initial accuracy, so this behaviour is expected and also consistent among different CSMs**: stronger regularization reduces the initial accuracy of the model because it penalizes sidechannel usage. The takeaway from this experiment is that _intervenability_ improves with stronger regularization, which is the _increase in accuracy_ after intervening on increasingly more concepts. This is because the less a CSM relies on its sidechannel, the more it must rely on its concepts. **Eventually, after enough interventions, the model trained with stronger regularization catches up in accuracy (e.g. Figure 4a, 4b).**
>
> &nbsp;
>
> > CSMs introduce a sidechannel to enhance task accuracy, thereby trading some interpretability for higher accuracy. Why would one intentionally reduce accuracy to regain interpretability in such a model — wouldn’t it be more straightforward to simply use a standard CBM instead? Could you clarify what benefits are gained from applying SIS regularization?
>
> **We show that SIS regularization can improve interpretability significantly without hurting any accuracy (lines 402-405), because current CSM optimization schemes do not consider interpretability.** This means it is generally always beneficial to use some SIS regularization. Compared to a standard CBM, typically, the CSM will be significantly more accurate (but less interpretable).
>
> **In case the resulting CSM is still not interpretable enough for the use case the human has in mind, they can use even more SIS regularization to start sacrificing accuracy for interpretability.** By controlling the strength of the regularization, the human can move the CSM within its accuracy-interpretability trade-off as much as required towards a standard CBM (SIS=100%) (e.g. Figure 2a). Whether this is beneficial depends on the exact use case (how much interpretability is required).
>
> &nbsp;
>
> > In line 240, “use” should be corrected to “us”.
>
> Thank you, we will correct this typo.
>
> &nbsp;
>
> [1] Mahinpei et al., 2021. Promises and pitfalls of black-box concept learning models.
>
> [2] Zarlenga et al., 2022. Concept embedding models.
>
> [3] Barbiero et al., 2023. Interpretable neural-symbolic concept reasoning.
>
> [4] Debot et al., 2024. Interpretable concept-based memory reasoning.
>
> [5] Linear Residual Model, an intermediate between [1] and Concept Bottleneck Models with Additional Unsupervised Concepts, Sawada et al. 2022.

---

> > ### Comment · Reviewer_XPHp · 2025-11-27
> >
> > Thank you for taking the time to carefully read my review and for writing a rebuttal that addresses my earlier concerns. I would appreciate it if you could further clarify the apparent tension between your two statements: “SIS regularization can improve interpretability significantly without hurting any accuracy” and “adding SIS regularization may harm initial accuracy”. It would be helpful to understand more precisely how these two claims differ and under what conditions each applies.

---

> > > ### Author Response · Authors · 2025-11-27
> > >
> > > > I would appreciate it if you could further clarify the apparent tension between your two statements: “SIS regularization can improve interpretability significantly without hurting any accuracy” and “adding SIS regularization may harm initial accuracy”. It would be helpful to understand more precisely how these two claims differ and under what conditions each applies.
> > >
> > > A small amount of SIS regularization may improve significantly without hurting accuracy (Figures 2a, 2b, 2e). This is because the current training objectives of CSMs only optimize for accuracy and not for interpretability. This means they may duplicate e.g. concept-related information in the sidechannel and rely entirely on the sidechannel, not on the concepts. **With a small amount of SIS regularization, the model is encouraged to not duplicate the concepts into the sidechannel**, which means they will use the concepts as much as possible and then use the sidechannel in case the concepts are insufficient. **This does not harm accuracy and improves interpretability.**
> > >
> > > **With a larger amount of SIS regularization, it will encourage the model to use the sidechannel less, _even if there is useful information in it that is not captured by the concepts_. This _does_ harm accuracy**, as less information is available to predict the task, **and it will improve interpretability even more**. For instance, in MNIST-Addition (Figure 2e, lines 419-423), all relevant information is captured by the concepts, so expressive CSMs can achieve perfect representation interpretability without a decrease in accuracy using SIS regularization. In CelebA (Figure 2a, 2b), not all relevant information is captured by the concepts, so both phenomena apply.
> > >
> > > More generally, **this pattern is consistent with the well-known behaviour of regularization in machine learning**. Moderate regularization can improve generalization and interpretability by discouraging the model from relying on spurious or redundant degrees of freedom, thereby pushing it toward solutions that better reflect the intended structure of the problem. However, as the strength of a regularizer increases, it inevitably constrains the hypothesis space more aggressively. When the imposed constraints begin to exclude functions that genuinely capture task-relevant information, performance degrades. Thus, the two statements are not in conflict: mild regularization operates in a beneficial regime where it removes undesirable shortcuts without limiting the model’s effective capacity, whereas strong regularization pushes the model into an under-fit regime where accuracy is sacrificed for greater interpretability.

---

### Official Review · Reviewer_vR2k · 2025-10-28

**Soundness:** 3
**Presentation:** 1
**Contribution:** 2
**Rating:** 4
**Confidence:** 5

**Summary:**

This paper is a follow-up research on concept bottleneck models. First, it unified CSMs and divided the interpretability into functional and representation interpretability. Based on this, this paper proposes a metric, SIS, that measures interpretability and can be used as a training objective. The paper conducts numerous experiments to demonstrate that SIS is highly useful and thoroughly discusses its interpretability.

**Strengths:**

- Though I give a negative rating, I really appreciate the unification of interpretability using functional and representation interpretability.
- The discussion on interpretability is insightful and reasonable.

**Weaknesses:**

- My primary concern is that the main contributions in this paper highly overlap with the existing work DCBM [1], which is not referenced. Specifically, they also design a residual branch and theoretically prove the trade-off between concept and label accuracy. More importantly, they propose a new metric, the concept contribution score (CCS), which is similar to SIS at a high level.
- Apart from the unification of interpretability and the interpretability discussion, the technique itself appears to be weak, which only introduces a metric and trains it as a regularization.
- It seems that functional and representation interpretability cannot cover all situations. For example, the joint concept bottleneck model meets both functional and representation interpretability; however, it still suffers from information leakage due to the continuous values in the representation. Do you consider this case?
- A minor concern is that the writing needs to be improved. For example, the contents in Section 3.1 could be organized as mathematical assumptions and definitions rather than plain text. The methodology section (Section 3) and the experiment section (Section 5.2) are divided into too many subsections or subsubsections. Though it demonstrates a high workload, it might fail to focus on the most essential parts.
- This is not a weakness, but the abbreviations in this paper are different from those in most existing works. For example, CBM typically represents the concept bottleneck model in the literature, but it is CBNM in this paper. Instead, the authors denote CBM as concept-based models. This might be confusing for researchers familiar with Concept bottleneck models, though these abbreviations are clearly defined at the beginning.

[1] The Decoupling Concept Bottleneck Model.

**Questions:**

See weaknesses.

---

> ### Author Response · Authors · 2025-11-21
> **Rebuttal (Part 1)**
>
> We thank the reviewer for taking the time to read our paper, and are glad the reviewer appreciates our unification of interpretability using functional and representation interpretability, and considers our discussion on the matter insightful and reasonable.
>
> &nbsp;
>
> > My primary concern is that the main contributions in this paper highly overlap with the existing work DCBM [1], which is not referenced. Specifically, they also design a residual branch and theoretically prove the trade-off between concept and label accuracy. More importantly, they propose a new metric, the concept contribution score (CCS), which is similar to SIS at a high level.
>
> Thank you for pointing out this work. **We will add this work to our Related Work section. Still, we believe very important differences exist, so it does not undermine our novelty:**
> - **DCBM is a concrete CSM with an embedding as sidechannel, while our work introduces CSM-agnostic methods and studies existing CSMs.** DCBM has a 'factorized' task predictor (similar to [2, 3]) where the task is the sum of the sidechannel's contribution and the concepts' contribution to the task. In contrast, our work is entirely model-agnostic: we do not make any assumptions about the sidechannel or task predictor.
> - **DCBM has a focus on e.g. concept leakage, while our contribution is orthogonal to this.** Their theoretically proven trade-off between concept and label accuracy assumes joint training of CBMs and is related to concept leakage. We do not consider concept leakage, and in our experiments we take measures to avoid it (e.g. no joint training, and binary concepts rather than continuous ones).
> - **DCBM's metric (CCS) does not measure representation interpretability, while ours (SIS) does.** CCS measures a relative difference between the logits of DCBM's 'standard' prediction and 'concept-based prediction', while SIS is instead accuracy-based, like representation interpretability. **CCS may be high even when representation interpretability is low**, specifically when the predictions are close to the decision boundary (similar to the metric of [4], see Appendix C.2 for a discussion). **This is not the case for SIS.** Moreover, CCS assumes a factorizable task predictor, which most CSMs don't have.
> - **DCBM's regularization to improve interpretability requires a factorizable task predictor (similar to e.g. [2]) and is thus not applicable to all existing CSMs, while ours is CSM-agnostic.**
>
> &nbsp;
>
> > Apart from the unification of interpretability and the interpretability discussion, the technique itself appears to be weak, which only introduces a metric and trains it as a regularization.
>
> **We also contribute the unification of all existing CSMs, which is what makes the definition of our CSM-agnostic metric and regularization possible.** Our metric and regularization are the first to measure and optimize for representation interpretability, which allow CSM practitioners to evaluate and optimize their models for interpretability instead of only accuracy. The metric and regularization allowed us to discover interesting phenomena about CSMs, such as:
> - Under existing training schemes, CSMs rely on the sidechannel even when completely unnecessary (lines 402-407).
> - More functionally interpretable CSMs rely on the sidechannel to become more expressive, meaning they have a trade-off between functional and representation interpretability (lines 303-310, lines 412-417).
> - A small amount of SIS regularization may significantly improve interpretability with no cost in accuracy (lines 402-405, lines 409-411).
> - By using a larger amount of SIS regularization, the human can sacrifice accuracy to become even more interpretable (lines 405-407).
>
> &nbsp;
>
> > It seems that functional and representation interpretability cannot cover all situations. For example, the joint concept bottleneck model meets both functional and representation interpretability; however, it still suffers from information leakage due to the continuous values in the representation. Do you consider this case?
>
> **While concept leakage is certainly an interesting problem, it is not the topic of our paper.** We assume that the concepts are correctly aligned and no leakage occurs (line 149, footnote 2). If leakage occurs, the concepts can indeed not be considered interpretable. **Our work is complementary to existing works on concept leakage, and we avoid the leakage problem in our experiments** by (1) not training the models jointly and (2) modelling the concepts as binary variables as opposed to continuous ones (Equation 5, line 804).

---

> ### Author Response · Authors · 2025-11-21
> **Rebuttal (Part 2)**
>
> > The contents in Section 3.1 could be organized as mathematical assumptions and definitions rather than plain text. The methodology section (Section 3) and the experiment section (Section 5.2) are divided into too many subsections or subsubsections.
>
> Thank you for this suggestion. We will re-organize our paragraphs on representation and functional interpretability as definitions, assumptions and propositions, and merge Sections 3.4 and 3.5.
>
> &nbsp;
>
> [1] The Decoupling Concept Bottleneck Model.
>
> [2] Incremental residual concept bottleneck models. Shang et al.
>
> [3] Post-hoc concept bottleneck models. Yuksekgonul et al.
>
> [4] Addressing leakage in concept bottleneck models. Havasi et al.

---

> > ### Comment · Reviewer_vR2k · 2025-11-26
> >
> > Thank you for providing the detailed answers. I appreciate this clarification and acknowledge your illustration of 1) the different main contributions compared to DCBM and 2) the exclusion of concept leakage. However, I think the paper requires further polishing before acceptance, including a more precise, more complete definition that covers all cases in CBNM and additional, more thorough experiments (if possible). Moreover, though I point out that the difference in abbreviations is not a weakness, I also hope the authors could update them in the paper to make it more straightforward. Based on all the above, I maintain the current rating in this stage.

---

> > > ### Author Response · Authors · 2025-11-27
> > >
> > > > Thank you for providing the detailed answers. I appreciate this clarification and acknowledge your illustration of 1) the different main contributions compared to DCBM and 2) the exclusion of concept leakage. However, I think the paper requires further polishing before acceptance, including a more precise, more complete definition that covers all cases in CBNM and additional, more thorough experiments (if possible). Moreover, though I point out that the difference in abbreviations is not a weakness, I also hope the authors could update them in the paper to make it more straightforward. Based on all the above, I maintain the current rating in this stage.
> > >
> > > We appreciate that we resolved the reviewer's main concern.
> > >
> > > We would like to remark the following:
> > > - In the revised version, **concept alignment is part of our definition of representation interpretability** (through Assumption 3.1), and therefore leakage is as well. We believe all cases of CSMs and CBNMs are therefore captured. Moreover, leakage cannot occur in our experiments due to our taken measures.
> > > - **We have added experiments on a new dataset (CUB) that confirm our results** (Figure 10).
> > >
> > > We kindly ask the reviewer to mention whether any concerns remain, and if not, to update their review based on our rebuttal.

---

> > > > ### Comment · Reviewer_vR2k · 2025-11-28
> > > >
> > > > Sorry for overlooking your supplementary experiments in the appendix. I have now reviewed your newly added empirical results, and **I am leaning toward raising the rating to 6**.
> > > >
> > > > Since score revisions are currently not accessible in the system, I hope the AC could take this into account. I will update the score correspondingly if the system is reopened.
> > > >
> > > > Finally, the following is some further discussion:
> > > >
> > > > 1. The author said `leakage cannot occur in our experiments due to our taken measures.' - I fully agree with it since the authors did not use a joint training strategy. However, I believe it would be interesting and important if your theoretical definition could also cover the leakage situation. Note that I am not forcing the authors to consider leakage at this time, but the authors could mention it as a future direction, maybe.
> > > >
> > > > 2. The authors did not respond to my comments on abbreviations. I hope the authors could 1) illustrate the reasons why you would not revise these abbreviations, or 2) acknowledge that you will revise them.

---

> > > > > ### Author Response · Authors · 2025-12-03
> > > > >
> > > > > Our intention was to use the abbreviation CBNM to be able to distinguish between Concept Bottleneck Models and the more general Concept-Based Models. But we agree with the reviewer that this will clash with the more common use of CBM for bottleneck models. Therefore, **we will change the abbreviations**, using the standard CBM for Concept Bottleneck Models (as the reviewer suggest) and we will introduce CBaM for Concept-Based Models.
> > > > >
> > > > > We will definitely **add the important problem of concept leakage as a direction for future work** in our conclusion.

---

### Official Review · Reviewer_ppDy · 2025-10-30

**Soundness:** 2
**Presentation:** 3
**Contribution:** 2
**Rating:** 2
**Confidence:** 3

**Summary:**

This paper addresses representation interpretability in Concept Sidechannel Models (CSMs) by proposing a unified probabilistic framework, introducing the Sidechannel Independence Score (SIS), and demonstrating how SIS regularization controls the accuracy-interpretability trade-off. Authors also analyze how the expressivity of the predictor and the reliance of the side channel jointly shape interpretability, revealing inherent trade-offs across different CSM architectures.

**Strengths:**

Strengths:

1.  The authors clearly articulate that CSMs can rely heavily on side channels (even when unnecessary)—this is a valuable empirical finding.

2.  The distinction between representation interpretability and functional interpretability is well-articulated and addresses a genuine gap in the literature. This separation clarifies previously muddled discussions about what makes concept-based models interpretable.

**Weaknesses:**

Weaknesses:

1. The paper defines representation interpretability as binary. Predictions must be "derived exclusively from interpretable units" (lines 146-149). Yet treats it as continuous, claiming CSMs are "partially representation interpretable" (lines 152-153). This contradiction undermines the value proposition: at SIS=100%, CSMs reduce to expensive CBNMs; at SIS<100%, some predictions are completely uninterpretable by the paper's own definition. The paper does not justify why practitioners would prefer models with unpredictable interpretability over either fully interpretable CBNMs (accepting accuracy loss) or black-box models (abandoning interpretability claims entirely).

2. Limited Scope of experiments: The paper considers only vision datasets and lacks a comprehensive evaluation.

3. The paper does not include a discussion on how the different CSM models differ from one another (Lines 366–371). Providing a more detailed explanation of these models would help readers better understand and interpret the results.

4. The paper does not include a clear definition of intervenability, which reduces the overall readability and clarity of the work.

5. The paper lacks the discussion on computational overhead of the SIS regularization.

**Questions:**

Questions to Authors:

1. Binary vs. Continuous Interpretability Contradiction: Your definition explicitly states predictions must be "derived exclusively from interpretable units" to be representation interpretable (lines 151-153), which is a binary property. However, you then claim CSMs are "partially representation interpretable" based on SIS scores. Can you clarify, doesn't any SIS<100% make the model unreliable for interpretability requirements for safety-critical applications?

2. For practitioners prioritizing interpretability, what is the concrete advantage of a CSM with SIS=60-80% over the two obvious alternatives, CBNMs with SIS=100% (fully interpretable)  or black box models (no interpretability, post-hoc explanations)?

3. How does SIS regularization affect the optimization landscape?

---

> ### Author Response · Authors · 2025-11-21
> **Rebuttal (Part 1)**
>
> We thank the reviewer for taking the time to read our paper. We appreciate that reviewer finds our empirical findings valuable, and our distinction between representation interpretability and functional interpretability well-articulated and a genuine gap in the literature that clarifies previously muddled discussions about interpretability.
>
> &nbsp;
>
> > The paper considers only vision datasets and lacks a comprehensive evaluation.
>
> **We evaluate a considerable number of CSMs ([1-5]) on two standard datasets that together are representative for most concept-based datasets:** CelebA is a setting where concepts are insufficient for perfect prediction with an inexpressive task, while MNIST-Addition has sufficient concepts with an expressive task. These datasets allow us to discover and show many phenomena regarding the use of CSMs in these two prominent settings (e.g. the use of the sidechannel to increase expressivity in some CSMs).
>
> **Following your suggestion, we've now added _a new third dataset (CUB)_ which confirms our results.** As CUB's task is multiclass classification, we cannot use CMR and DCR; only CRM, LRM and CEM. We attach the pareto curve for seed 1 (similar to Figure 2 in the paper). We've added the other seeds in the appendix.
>
> &nbsp;
>
> > The paper defines representation interpretability as binary: predictions must be "derived exclusively from interpretable units" (lines 146-149). Yet treats it as continuous, claiming CSMs are "partially representation interpretable" (lines 152-153). This seems a contradiction: can you clarify this?
>
> We define the representation interpretability _of a prediction_ as binary (lines 146-149), and mean that a CSM is more representation interpretable the more often its predictions are representation interpretable. **We will clarify this in the text.**
>
> &nbsp;
>
> > Doesn't any SIS<100% make the model unreliable for interpretability requirements for safety-critical applications?
>
> For safety-critical applications, a fully interpretable model may be necessary. **In such cases, the human should indeed use a concept bottleneck model**, or a CSM with SIS=100% (which is practically equivalent). **Over recent years, CSMs have been developed not to provide a fully interpretable model**, but a model that is as accurate as possible and provides interpretability on top.
>
> &nbsp;
>
> > At SIS=100%, CSMs reduce to CBNMs; at SIS<100%, some predictions are completely uninterpretable. The paper does not justify why practitioners would prefer partially interpretable models over either fully interpretable CBNMs (SIS=100%) (accepting accuracy loss) or black-box models (no interpretability).
>
> **This is related to why CSMs have been developed over recent years.** The central idea behind the development of all existing CSMs is primarily to achieve (near) black-box accuracy, and secondarily to be as interpretable as possible. With our work, we can measure this interpretability and even optimize for it, which was not the case before.
>
> **Interpretability is not black and white; there are degrees of interpretability.** How much interpretability is desired depends on the human and the use case of the model. This is intuitive to see with functional interpretability: generally, people consider a shallow decision tree with depth 3 interpretable, which does not mean that depth 4 is entirely uninterpretable for them; it is just _less interpretable_ (but may be more accurate). The same applies to representation interpretability (but contrary to functional interpretability, we can measure this objectively). It depends on the human: how much they value interpretability versus accuracy. **We will state this in the Background section.**
>
> &nbsp;
>
> > The paper does not include a discussion on how the different CSM models differ from one another (Lines 366–371). Providing a more detailed explanation of these models would help readers better understand and interpret the results.
>
> **We provide a detailed explanation of 6 existing CSMs in Appendix A, and in the main text we explain two of them at lines 195-208.** At lines 369-371, we mention the expressivity of each CSM and whether they are functionally interpretable, which is necessary for interpreting the results that follow.
>
> &nbsp;
>
> > The paper does not include a clear definition of intervenability, which reduces the overall readability and clarity of the work.
>
> **Intervenability is a well-known idea in the concept-based community, and is explained in our Related Work section (Section 4, line 325).** By increasingly replacing concept predictions with their ground truth label at test-time, we want downstream task accuracy to increase as much as possible. This simulates human expert interaction.

---

> > ### Comment · Reviewer_ppDy · 2025-11-27
> > **Rebuttal Response**
> >
> > I thank the authors for their detailed response. The authors have addressed my queries. In view of the author's clarifying comments and reviewer comments. I am raising the score to 5.

---

> ### Author Response · Authors · 2025-11-21
> **Rebuttal (Part 2)**
>
> > The paper lacks the discussion on computational overhead of the SIS regularization.
>
> **The SIS regularization requires a single forward pass of the model** (when using the learnable prior) **and computing a KL divergence** between the original forward pass and this additional one (see the equation in Section 3.5), so this is cheap.
>
> &nbsp;
>
> > How does SIS regularization affect the optimization landscape?
>
> **The SIS regularization can be analysed in terms of a prior regularization.** The two CSM modes represent two variational distributions: a posterior p(y|c,x) and a prior p(y|c).
>
> If we focus on the standard CBM likelihood term (p(y_data|c,x)), the role of SIS (as any other prior regularization) is to reshape the likelihood surface by adding attraction basins around predictions consistent with the prior. The SIS term alters the gradient such that predictions that deviate from the prior still incur a cost even if they fit the data perfectly.
>
> From an intuitive geometric perspective, peaks of the likelihood surface are “pulled” toward regions where model outputs resemble the prior. Spurious local maxima may disappear if they violate the prior. New local maxima can appear if the prior has strong structure. The weight balances these two behaviours.
>
> This is exactly what happens in e.g. a standard variational auto-encoder: the KL term regularizes the posterior toward the prior, reshaping the learning surface [6]. **Using SIS regularization, the CSM's interpretability improves, because the prior p(y|c) is representation interpretable, while the posterior p(y|c,x) not necessarily. By pulling the prior and posterior closer to each other, the CSM becomes more interpretable.**
>
> &nbsp;
>
> [1] Mahinpei et al., 2021. Promises and pitfalls of black-box concept learning models.
>
> [2] Zarlenga et al., 2022. Concept embedding models.
>
> [3] Barbiero et al., 2023. Interpretable neural-symbolic concept reasoning.
>
> [4] Debot et al., 2024. Interpretable concept-based memory reasoning.
>
> [5] Linear Residual Model, an intermediate between [1] and Concept Bottleneck Models with Additional Unsupervised Concepts, Sawada et al. 2022.
>
> [6] Kingma & Welling, 2013. Auto-encoding variational bayes.

---

### Author Response · Authors · 2025-12-03
**Summary for AC**

Dear AC,

We want to give a concise summary of the reviews, our rebuttal and our corresponding changes. We also want to remark the following:


| **Reviewer** |  **Initial score** (before the rebuttal) | **Comment after the rebuttal** |
| -------- | ------- | -------- |
| **ppDy** |  **2** | “I am raising the score to **5**.“ [*We know 5 is not an admissible score.*] |
| **vR2k**   |   **4** | “I am leaning toward raising the rating to **6**.**” |
| **XPHp**  |   **4** | “Thank you for [...] and for writing a rebuttal that addresses my earlier concerns” [*and asked another clarification before editing became disabled.*]  |


### Common points

- *Reviewer ppDy and XPHp asked for **more datasets**.* As a response, we **added a new dataset** (CUB), and pointed out that we already used two representative concept-based datasets (CelebA, MNIST-Addition), and evaluate 5 state-of-the-art models on them.


### Reviewer ppDy

The reviewer primarily asked for clarifications, and our rebuttal mostly consisted of directing them sections of the paper where the points were already addressed. We also provided an experiment with an additional dataset. The reviewer said that we addressed all their queries.

- *They found a contradiction in our writing*, which we addressed.
- *They asked a question related to **our research area (CSMs)** in general: why a human would use (the already existing) CSMs?* We explained the reasoning behind their development over recent years.
- *They asked for a **discussion about how existing CSMs differ** from each other.* We remarked that **we already discuss many** of them in the paper (2 in the main text and 6 in the appendix).
- *They asked to **explain 'intervenability'** in the text.* We remarked that we **already explained** this in the text.
- *They asked for a **discussion on computational overhead** of our regularization.* We explained that it requires a single additional forward pass of the model and a KL divergence.
- *They asked how our regularization affects the **optimization landscape**.* We explained this in terms of a **prior regularization**.


### Reviewer vR2k

The reviewer had a small number of questions regarding related work and a possible direction for future work. We added a citation and expanded the Related Work section. The reviewer indicated that our clarifications resolved their concerns.

- *Their primary concern was an **unreferenced paper** they believed had **high overlap** with ours.* We pointed out some **very important differences** (_to which the reviewer agreed_), and added this explanation and reference in our Related Work section.
- *They mentioned that our metric and regularization appear to be technically weak.* We remarked that we also contributed (1) distinguishing and defining two types of interpretability in CSMs, (2) a unification of all existing CSMs, and that our metric and regularization (which are model-agnostic) are only possible because of (1) and (2).
- *The reviewer asked about how **concept leakage** fits in our work.* We replied it is **orthogonal to our work**, and that we have an explicitly-made assumption about it in our text.



### Reviewer XPHp

The reviewer focused on the theoretical guarantees of our metric and clarifications about experiments. The reviewer stated that our responses resolved their concerns.

- *They said our metric lacks **theoretical guarantees**.* We pointed out that our metric **_does_ provide theoretical guarantees**, as explained in the text.
- *They asked questions about some phenomena in our *intervenability* results*, which we explained (e.g. the difference between accuracy and intervenability).
- *They asked **why** one would want to **reduce accuracy to improve interpretability**.* We explained that (1) a small amount of our regularization significantly improves interpretability *without* hurting accuracy, and (2) if that is still not interpretable enough for the human's use case, they may want to sacrifice accuracy.

---

### Meta-Review · Area_Chair_hUsS · 2025-12-24

**Summary:**

This paper examines the fundamental trade-off between accuracy and interpretability in Concept Bottleneck Models (CBMs), a critical challenge in interpretable machine learning. Its primary contribution lies in the introduction of the Sidechannel Independence Score (SIS), a novel metric that quantifies the reliance of the CSMs (Concept Sidechannel Models, i.e. Concept Bottleneck Models endowed with a non-interpretable sidechannel) on the sidechannel, bypassing the interpretable concepts and therefore constituting a non-interpretable residual contributing to the prediction. The paper demonstrates that using the SIS as a regularization helps control the accuracy-interpretability trade-off, which provides a practical tool to mitigate reliance on the non-interpretable pathway.

Reviewers praised the paper for its clarity, for introducing the conceptual distinction between representational and functional interpretability, and articulating the trade-off of relying on the sidechannel over the intepretable concepts, while also empirically validating this trade-off.

In terms of novelty, the paper's contribution remains nuanced. While the SIS metric is indeed a novel formulation of the reliance on non-interpretable sidechannels, the reviewers point out that the paper is largely a follow-up methodological contribution aimed at refining and repositioning CSMs, building upon established frameworks like Decoupling Concept Bottleneck Model. The rebuttal successfully clarifies some differences and generalization over that work, though the core overlap in terms of proposing to measure and control dependence on concepts in CBMs undermines claims of groundbreaking innovation.

A more pressing limitation is the narrow scope of the domain within which the paper is formulated. On one hand, reviewers pointed to a limited empirical scope, with experiments confined to vision tasks of relatively small scale, which limits the generalizability of the findings.
Although the rebuttals provide an analysis on an additional dataset (CUB-200), the dataset is still a vision task. Moreover, it is notable that the experiments on the CUB-200 dataset (a staple in the interpretable machine learning and explainability literature) do not reach accuracies competitive with the state-of-the-art explainability models such as Concept-Centric Transformer and ConceptTransformer, even when SIS is lowered to levels that significantly compromise interpretability.

This raises broader concerns regarding the significance and practical utility of the proposed approach in real-world applications where both interpretability and competitive accuracy are desired.
While the proposed Sidechannel Independence Score (SIS) is arguably a valuable contribution to interpretable machine learning, allowing for a quantification of the reliance of Concept Bottleneck Models (CBMs) on non-interpretable sidechannels, the practical impact of the findings may be limited by the fact that, as observed in the reviews, any amount of reliance on the sidechannel limits the interpretability of the model in real-world applications. This is not a limitation of the current paper per se, but rather a broader challenge in the field of interpretable machine learning, and Concept Sidechannel Models in particular.

To summarize, while the rebuttals addressed some of the concerns raised by reviewers, they did not fully alleviate the doubts regarding the novelty, generalizability, and practical impact of the work. In particular, the limited scope around understanding and controlling a specific subclass of interpretability models (CSMs), the conceptual concerns around the implication of giving up on interpretability for any amount of sidechannel reliance, and the limited empirical validation on vision tasks only, all contribute to a sense that the work, while solid, may need repositioning in order to provide a wider impact in the field.

**Reviewer Concerns:**

* Addressed in the rebuttal:
  - lack of differentiation with respect to established models like Decoupling Concept Bottleneck Model raised by reviewers has been partially addressed in the rebuttal by clarifying the generalization and differences of the proposed framework
  - theoretical guarantees: initial misunderstanding about the theoretical guarantees of SIS regularization have been clarified in the rebuttal, specifying that the SIS metric can provide an upper bound on sidechannel reliance under certain assumptions

* Not addressed in the rebuttal:
  - limited novelty: the paper is largely a follow-up methodological contribution aimed at refining and repositioning CSMs, building upon established frameworks
  - limited empirical evaluation: paper only examines vision tasks of relatively small scale, which limits the generalizability of the findings
  - relation of other established concept-based models (e.g., Concept-Centric Transformer, ConceptTransformer) not completely discussed

**Reviewer Scores:**

| Reviewer | initial score | predicted final score |
|---:|---:|---:|
| ppDy | 2 | 4 (reviewer indicated they will raise score to 5) |
| vR2k | 2 | 6 (indicated by reviewer in discussion) |
| XPHp | 4 | 4 |

---

### Decision · Program_Chairs · 2026-01-26

Reject